# Determinants of trafficking, conduction, and disease within a K⁺ channel revealed through multiparametric deep mutational scanning

Willow Coyote-Maestas[1†], David Nedrud[1], Yungui He[2], Daniel Schmidt[2]*

[1]Department of Biochemistry, Molecular Biology and Biophysics, University of Minnesota, Minneapolis, United States; [2]Department of Genetics, Cell Biology and Development, University of Minnesota, Minneapolis, United States

*For correspondence:
schmida@umn.edu

Present address: [†]Department of Bioengineering and Therapeutic Science and Quantitative Biosciences Institute, University of California, San Francisco, United States

Competing interest: The authors declare that no competing interests exist.

**Abstract** A long-standing goal in protein science and clinical genetics is to develop quantitative models of sequence, structure, and function relationships to understand how mutations cause disease. Deep mutational scanning (DMS) is a promising strategy to map how amino acids contribute to protein structure and function and to advance clinical variant interpretation. Here, we introduce 7429 single-residue missense mutations into the inward rectifier K⁺ channel Kir2.1 and determine how this affects folding, assembly, and trafficking, as well as regulation by allosteric ligands and ion conduction. Our data provide high-resolution information on a cotranslationally folded biogenic unit, trafficking and quality control signals, and segregated roles of different structural elements in fold stability and function. We show that Kir2.1 surface trafficking mutants are underrepresented in variant effect databases, which has implications for clinical practice. By comparing fitness scores with expert-reviewed variant effects, we can predict the pathogenicity of 'variants of unknown significance' and disease mechanisms of known pathogenic mutations. Our study in Kir2.1 provides a blueprint for how multiparametric DMS can help us understand the mechanistic basis of genetic disorders and the structure–function relationships of proteins.

## Editor's evaluation

This manuscript details a new approach to systematically interrogating the relations between the amino acid sequence of ion channels, and their stability, subcellular localization, and function. By applying this approach to an inward-rectifier K⁺-channel, the authors uncover intriguing structural relations between channel regions likely involved in protein stability and those responsible for functional dynamics. Their analysis also offers predictions for the potential pathogenicity of channel variants of unknown significance found in the human population.

## Introduction

Inward rectifier K⁺ (Kir) channels play a central role in physiology by setting, maintaining, and regulating a cell's resting membrane potential (RMP) (*Hibino et al., 2010*). Misregulated Kir's cause neurological disorders (alcohol and opiate addiction; *Bodhinathan and Slesinger, 2013*; *Marron Fernandez de Velasco et al., 2015*; *Torrecilla et al., 2002*), Down's syndrome (*Rachidi and Lopes, 2007*), epilepsy (*Köhling and Wolfart, 2016*), Parkinson's (*Zhang et al., 2020*), cardiac disorders (long QT syndrome; *Moss and Kass, 2005*), hereditary renal diseases (*Flagg et al., 1999*), and diabetes (*Flanagan et al., 2009*; *Hattersley and Ashcroft, 2005*; *Marthinet et al., 2005*).

**Figure 1.** Deep mutational scanning (DMS) to improve clinical variant prediction. (**A**) Clinically observed variation in Kir2.1 and other Kir and their interpretation of pathogenicity. (**B**) Schematic summarizing the various processes that must work correctly for Kir2.1 to fulfill its role in cellular physiology. (**C**) Schematic summarizing DMS and multiparametric assessment of phenotype. Mutations are introduced into Kir2.1 using SPINE. A stable single-copy insertion library is generated by BxBI-mediated recombination in HEK293T. Cells are sorted based on channel surface expression or function (K⁺ conductance) as determined by antibody labeling of an extracellular FLAG-tag or voltage-sensitive dye, respectively. Genotypes of each sorted cell population are recovered by NextGen Sequencing (NGS).

The online version of this article includes the following figure supplement(s) for figure 1:

**Figure supplement 1.** Kir2.1 variants reported in ClinVar and gnomAD.

**Figure supplement 2.** Stable library cell line quality control.

**Figure supplement 3.** Kir2.1 surface expression assay.

**Figure supplement 4.** Kir2.1 functional fitness assay gating scheme.

While some mechanisms that underlie disease-causing Kir variants involve aberrant function (gating, ion permeation, ligand binding; *Abraham et al., 1999*) or transcript processing (*Vuong et al., 2016*), others are linked to trafficking defects. For example, over half the missense mutations investigated in Kir1.1 affected surface expression, often caused by proteosomal degradation of folding-deficient or mistrafficked channels (*O'Donnell et al., 2017*; *Peters et al., 2003*). A comparison of mutation hotspots in the Kir C-terminal domain (CTD) revealed that ~60% of variants could be linked to impaired surface expression (*Fallen et al., 2009*; *Zangerl-Plessl et al., 2019*). Mutations linked to neonatal diabetes (which is caused by gain-of-function mutations) and congenital hyperinsulinism affect Kir6.2 surface expression to varying degrees (*Flanagan et al., 2009*; *Hattersley and Ashcroft, 2005*; *Marthinet et al., 2005*; *Lin et al., 2003*). Beyond known trafficking signals (e.g., ER and Golgi-export signal sequences; *Li et al., 2016*; *Ma et al., 2001*; *Ma et al., 2002*; *Ma et al., 2011*; *Stock-klausner et al., 2001*; *Zerangue et al., 1999*), mutations along the entire Kir primary sequence can disrupt surface expression (*Zangerl-Plessl et al., 2019*). Several additional factors control surface expression, such as protein stability or interactions with trafficking partners and complexes that stabilize channels in the membrane (*Ma et al., 2001*; *Ma et al., 2002*; *Leonoudakis et al., 2004*).

The apparent prevalence of Kir trafficking phenotypes has a significant caveat: phenotypes are experimentally determined for only a small fraction of possible or clinically observed variants. Of the 8113 possible missense mutations for *KCNJ2*/Kir2.1, only 252 are reported in ClinVar – a database of clinically observed variation and phenotypes (*Landrum et al., 2018*) – and gnomAD – a database

of human variation from exome and whole-genome sequences (*Karczewski et al., 2020*; *Figure 1A*, *Figure 1—figure supplement 1*, *Supplementary file 1*). Most studies focus on common natural variants observed in human genomes or exomes. Rare genetic variants vastly outnumber common variants (*Tennessen et al., 2012*) and are implicated in a substantial portion of complex human disease (*McClellan and King, 2010*). Rarity means that clinical significance (benign or pathogenic) of most variants cannot be established using genome-wide association studies due to low statistical power for calling pathogenic variants. For *KCNJ2*/Kir2.1, ClinVar reports 162 missense mutations, but most (74%) have 'uncertain significance' or conflicting or no interpretation (*Figure 1A*). Computational algorithms (PolyPhen-2 [*Adzhubei et al., 2010*], SIFT [*Ng and Henikoff, 2003*], EVE [*Frazer et al., 2021*], etc.) are currently filling this gap by predicting the consequence of mutations. However, computational approaches work best in conserved regions of genes and cannot predict mechanism of pathogenicity.

Despite a central role for Kir trafficking and function in disease etiology, there are no large-scale studies that comprehensively study sequence determinants of Kir trafficking and function. Apart from an early study that identified functional Kir2.1 substitutions (*Minor et al., 1999*), previous studies focused on a small subset of common natural variants. Systematic studies are needed for a global picture of Kir trafficking and function. Beyond explaining disease, large-scale mutational studies can reveal intrinsic properties of proteins (*Atkinson et al., 2018*; *Coyote-Maestas et al., 2021*; *McLaughlin et al., 2012*), determine protein structures (*Schmiedel and Lehner, 2019*), and inform mechanistic models of protein structure–function relationships (*Matreyek et al., 2018*).

Here, we combine programmable mutagenesis library generation with multiparametric sequencing-based assays for surface expression and function of Kir2.1. Our data reveal distinct determinants for Kir trafficking, folding, and function. The phenotype of most clinically observed variants can be explained by impacting function, not surface expression. We also find further support for the hypothesis that a hierarchical organization of ion channels structure balances stability and flexibility for folding and function (*Coyote-Maestas et al., 2021*).

## Results

### A multiparametric deep mutational scan of Kir2.1

For Kir2.1 to fulfill its roles in cellular physiology, it must be ER targeted, folded, tetramerized, and surface trafficked (*Figure 1B*). Once trafficked to the cell surface, Kir2.1 must be sensitive to phosphatidylinositol-4,5-bisphosphate (PIP$_2$) (*Hibino et al., 2010*; *Hansen et al., 2011*), undergo gating-associated conformational transitions (pore opening and closing), and selectively conduct K$^+$.

To probe how Kir2.1 mutations affect these processes, we use a programmable deep mutagenesis approach (SPINE; *Coyote-Maestas et al., 2020*; *Nedrud et al., 2021*) and mutated residues 2–392 of mouse Kir2.1 (UniProt P35561) to every other amino acid (*Figure 1C*). We included synonymous mutations in 20 Kir2.1 positions as an internal standard to determine wildtype fitness. From this deep mutational scanning (DMS) library, we generated a stable single-copy library HEK293 cell line using BxBI-mediated recombination (*Matreyek et al., 2020*) for stringent genotype–phenotype linkage. Endogenous channel currents in HEK293 are negligible when compared to those of exogenous channel driven from a strong constitutive promoter (CAG) (*Varghese et al., 2006*). Sequencing of the stable library cell line indicates near-complete coverage and high redundancy: 90.2% of variants were detected; >89% of variants had a read count of greater than 20 (*Figure 1—figure supplement 2*).

Since mutations can affect any of the abovementioned processes, multiparametric assessment of phenotypes is required to develop a comprehensive mechanistic understanding of Kir2.1 variation. We focused on separating a mutation's effect on secretion, folding, and trafficking ('surface expression') and allosteric regulation and ion conductance ('function'). Considering the variant library size (391 × 19 = 7429 variants), assays that assess surface expression and function must be high-throughput. Fluorescently activated cell sorting (FACS) allows sorting of variant libraries based on protein phenotypes; sequencing can be used to assign a genotype to each sorted phenotype.

Mutations that interfere with secretion, folding, or trafficking will cause reduced Kir2.1 surface expression (*Bendahhou et al., 2003*; *Pegan et al., 2006*; *Tinker et al., 1996*). To measure Kir2.1 surface expression in cells, we inserted an epitope Flag-tag into the extracellular loop (T115; *Stocklausner et al., 2001*). We can determine surface expression scores for all mutants in the library by using FACS and NextGen Sequencing (NGS) on fluorescently antibody-labeled cells expressing the

Kir2.1 DMS library (*Figure 1—figure supplement 3*). Phenotypes can be linked to genotype by NGS. We have extensively used this method in FACS-based genotype–phenotype assays (*Coyote-Maestas et al., 2021*; *Coyote-Maestas et al., 2020*; *Coyote-Maestas et al., 2019*).

Functional Kir2.1 hyperpolarizes HEK293 cells by driving the RMP toward the reversal potential of $K^+$ (–80 mV for our conditions) while nonfunctional Kir2.1 variants do not affect RMP (–35 mV in HEK293). We can measure RMP with voltage-sensitive dyes, such as the FLIPR Blue, a concept that we and others demonstrated in optical Kir function assays (*Coyote-Maestas et al., 2019*; *Adams and Levin, 2012*). Changes in membrane voltage alter dye membrane partitioning, and therefore extinction coefficient and fluorescence. Hyperpolarized cells expressing functional Kir2.1 can be separated from more depolarized cells expressing nonfunctional Kir2.1 (e.g., the nonconducting V302M mutant; *Ma et al., 2007*) by FACS using FLIPR dye fluorescence (*Figure 1—figure supplement 4*). As with surface expression, phenotype can be linked to genotype by NGS.

## A global view of trafficking determinants

To learn how amino acids contribute to and mutations alter Kir2.1 trafficking and folding, we sorted and sequenced the Kir2.1 DMS library based on surface expression. Surface trafficking fitness was determined for 6898 Kir2.1 variants (93%). Of 521 missing variants (no data in either replicates), only 154 were also missing in the stable library cell line, suggesting that dropout is stochastic and not related to library construction or poor coverage in the original cell pool (*Figure 2—figure supplement 1*). Biological replicates for subpools were highly correlated (Pearson correlations 0.85–0.94, *Figure 2—figure supplement 2*) and read depth is excellent (greater than 30-fold at most positions, *Figure 2—figure supplements 3–4*). Surface expression fitness scores and standard errors were calculated using Enrich2 (*Rubin et al., 2017*). Enrich2 calculates log enrichment by fitting a weighted regression based on changes in mutation frequency across samples within an experiment. Positive scores represent enrichment (increased surface expression) greater than wildtype, while negative scores represent depletion relative to wildtype. In our assay, surface fitness follows a bimodal distribution with one population of mutations centered at wildtype fitness and the other population strongly decreasing surface expression (*Figure 2A*). Median and standard deviation for synonymous and missense mutations were –0.1 ± 0.4 and –0.17 ± 2.18, respectively (*Figure 2A*, *Figure 2—figure supplement 5*). 10% percentile tail ends differed by 4.1 and 0.94 log units at the low and high ends, respectively, which suggest that differences in surface trafficking are well-resolved.

As expected, mutations are highly deleterious within the Flag-tag used for surface labeling and known trafficking signals (*Figure 2B*). Substituting aromatic residues with charged residues strongly decreases surface expression within the $_{81}$WRWMLLLLF$_{88}$ motif that mediates interaction with Caveolin-3 (*Vaidyanathan et al., 2018*). Glutamates within the di-acidic ER export signal (*Ma et al., 2001*), $_{382}$FCYENE$_{387}$, are quite sensitive to mutations with the second glutamate being far more sensitive. Interestingly, negatively charged mutations, N386E or N386D, between the di-acidic motif increase surface expression (3.3 ± 0.76 and 3.9 ±1.2, respectively).

A Golgi export signal comprised of two residues patches at Kir2.1's N- and C-termini interacts with the AP1 adaptin complex to target Kir2.1 to clathrin-coated vesicles for Golgi to plasma membrane trafficking (*Ma et al., 2011*). Others have used alanine scans to identify sequence determinants for Golgi export. First, *Ma et al., 2011* defined the minimal signal motif comprised of hydrophobic residues (I328, W330, Y323) and a juxtaposed electrostatic interaction between R46 and E335. Later, *Li et al., 2016* included salt bridges between N-terminus (R44, R46, K50) and C-terminus (E301, E327), and hydrophobic residues (F203, L239, L240, F300). This structured trafficking signal is formed when the CTD has folded correctly to include a hydrophobic cleft, an adjacent cluster of basic residues, and a network of interactions between termini and adjacent subunits. Ma et al. suggested this predisposes Golgi exit on correct protein conformation; our comprehensive mutagenesis suggests this quality control mechanism may extend beyond previously identified sequences. It appears to involve all core β-sheets in the CTD (βD1, βH, βI, βJ, βK, βB2, βC, βG) and helices αG and αF (*Figure 2B and C*). Across CTD secondary structure elements, we find a distributed network of residues that, when mutated, result in reduced surface expression (*Figure 2D*). Changes to the packing of the hydrophobic core of the CTD will likely impact the Golgi export motif orientation. We hypothesize that the Golgi export signal is a reporter for folding of the entire CTD. Several lines of evidence support this hypothesis: IgG-like domains are a β-sheet sandwich with conserved hydrophobic residues at sheet interfaces

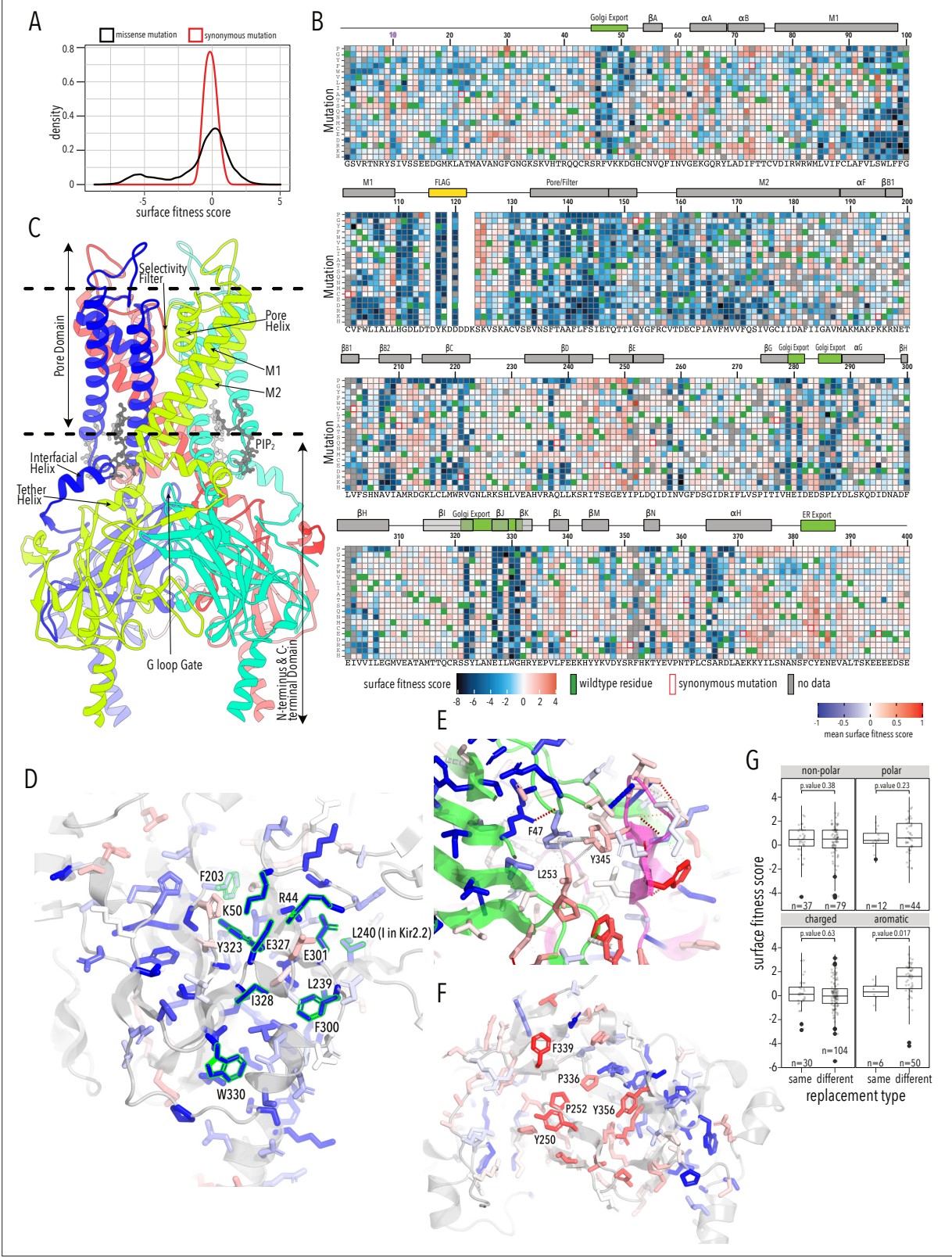

**Figure 2.** Surface expression fitness. (**A**) Distribution of surface expression fitness scores for synonymous mutations (red line) and missense mutations (black line). (**B**) Heatmap of Enrich2-calculated surface expression fitness scores expressed as a gradient from blue (below wildtype [WT] fitness) over white (same as WT fitness) to red (above WT fitness). WT residues are indicated by filled green boxes. Missing data is indicated by filled dark gray boxes. Synonymous mutations are indicated by red box outlines. (**C**) Cartoon of Kir2.2 (PDB 3SPI, 70% identity with Kir2.1). Monomer chains shown

*Figure 2 continued on next page*

*Figure 2 continued*

in different colors. Lipid bilayer boundaries are indicated by dashed lines. (**D**) Close-up of core β-sheet of the C-terminal domain (CTD) with mean surface expression fitness (averaged across all measured mutations for that position) mapped onto each residue as a gradient from blue over white to red, corresponding to below WT, same as WT, and above WT fitness, respectively. Residues that comprise the Golgi export signal are outlined green. Numbering corresponds to Kir2.1-Flag. (**E**) Interface close-up of neighboring subunits (green and magenta cartoons). Sidechains are colored by surface fitness score; same scale as in (**D**). (**F**) Close-up of βDE and βLM loop mean surface expression fitness. (**G**) Boxplots showing surface fitness in the βDE and βLM loop if the WT residues of a specific type (e.g., glutamate, charged) is mutated to another residue of the same type (e.g., aspartate) or a different type (e.g., tryptophan, aromatic). Median is marked with a thick line, the vertical length of the box represents the interquartile range (IQR), upper fence: 75th percentile +1.5× IQR, lower fence: 25th percentile −1.5× IQR, outlier points are shown as solid black circles. All data points are indicated by transparent dots. Only replacing aromatic residues significantly increases surface expression. Significance is tested using one-sided *t*-tests; number of observations (n) and p-values are indicated for each comparison.

The online version of this article includes the following figure supplement(s) for figure 2:

**Figure supplement 1.** Missing variants.

**Figure supplement 2.** Surface fitness assay biological replicates.

**Figure supplement 3.** Surface fitness assay read count depth.

**Figure supplement 4.** Surface assay read statistics and assay repeatability.

**Figure supplement 5.** Surface fitness assay standard error distribution.

that are crucial for fold stability (*Cota et al., 2001*; *Hamill et al., 2000*), computational mutagenesis in Kir1.1 showed these residues are crucial for fold stability, and experimental mutagenesis found that mutations strongly reduce inward rectifier surface expression (*Fallen et al., 2009*; *Koster et al., 1998*). Residues in this network belong to the same subunit; disrupting interactions between residues belonging to neighboring subunits (e.g., carbon–π and π–π interactions between F47, L253, Y345; *Figure 2B and E*) do not uniformly reduce surface expression. Although Kir is presumably already assembled into tetramers (*Deutsch, 2002*), our data suggests that the conformational checkpoint for Golgi export requires properly folded monomers not correctly assembled tetramers.

Sites enriched for beneficial or deleterious substitutions could point to additional mechanisms for Kir2.1 trafficking control. Along those lines, we observe that substitution of N- and C terminal residues (G2–G31, S377–S381) with polar mutations have neutral trafficking phenotypes, while aromatic or hydrophobic mutations tend to decrease surface expression. While we do not know the underlying mechanism, our data suggests that disordered N- and C-termini are important in targeting Kir2.1 to the cell surface.

Mutations along βB2/C(M211–K215), βDE(S242–P252), βK(R333, P336), βL(F339), βM(Y345), βN(Y356), and αF helix(K372–L376) tend to improve surface trafficking above WT especially when aromatic residues in the βDE and βLM loops are replaced with any non-aromatic amino acids (*Figure 2G*, *t*-test p-value 0.017). These sites form a patch on the outer face of the CTD domain where two subunits interdigitate (βDE from one subunit, the remaining elements from another subunit; *Figure 2F*). In this region, beneficial mutations have no clear preferences beyond non-aromatics. While no known trafficking signal matches, aromatic residues are often found in binding interfaces (*Vaidyanathan et al., 2018*; *Williams and Fukuda, 1990*). Perhaps mutations alter trafficking patterns by disrupting protein interactions (e.g., reduce ER retention, reduce forwarding to lysosomal recycling).

Overall, the agreement with earlier trafficking studies in Kir2.1 shows that our high-throughput surface expression assay works. This allows us to establish a global view of sequence/trafficking relationships in Kir2.1. Comprehensive mutagenesis allows us to validate and fill in the gaps on existing models of channel trafficking (*Li et al., 2016*; *Ma et al., 2011*) while discovering new potential trafficking determinants.

## A cotranslationally folded biogenic unit in Kir2.1

By examining structured regions sensitive to mutations, we can identify determinants of Kir2.1 folding and trafficking. In the pore domain, we find the lower halves of M1 and M2 helices are more tolerant to mutation compared to the upper halves. Residues whose sidechains project toward the pore cavity are particularly deleterious. At the base of the pore domain, mutations are highly deleterious within a cluster of hydrophobic residues (W81, L85, I178, F182, I187, M191) that form inter-subunit interactions between M1 and M2 helices. The top of the pore domain – a region comprised

of the upper halves of M1, M2, and pore helix – is extremely intolerant to substitutions. This includes the M1-pore helix loop (e.g., H110–E133), whose amino acid sidechains contribute to tight packing centered around a hydrophobic pocket formed by F103 (M1), V131 (M1-pore helix loop), F143 (pore helix), V158 (pore helix-M2 loop), and V169 (M2) (*Figure 3A*). The same region contains two highly conserved cysteines (C130, C162) that form a disulfide bond that is critical for folding but not function (K$^+$ conductance) (*Cho et al., 2000*; *Leyland et al., 1999*; *Tao et al., 2009*). Others found that P163, at M2's apex, is essential for efficient surface trafficking in Kir2.2 (*Dassau et al., 2011*). In agreement with these prior studies, all substitutions in these positions are deleterious to surface expression scores.

Glycines in the selectivity filter ($^{147}$T̲Q̲T̲T̲I̲G̲Y̲G̲$^{154}$ in Kir2.1; conserved residues are underlined) are absolutely conserved in all K$^+$ channels. Because glycine is achiral, it is the only canonical amino acid that can satisfy the unusual conformation of the selectivity filter main chain, which alternates between left-handed and right-handed α-helical dihedral angles (*Valiyaveetil et al., 2004*). Despite this strong conservation, the selectivity filter is significantly more tolerant to mutations compared to the preceding M1 and the following pore helix (*Figure 3B–D*, two-sample Kolmogorov–Smirnov test p-value 1.918e-13). Substitutions of either T150 (which occupies unfavorable backbone torsion angle conformation in the Kir2.2. crystal structure; *Hansen et al., 2011*; *Tao et al., 2009*) or Y153 (which forms π–π interaction to a neighboring subunit) in the selectivity filter with glycine do not decrease surface expression. This suggests that the peculiar conformation of the selectivity filter is not required for channel biogenesis.

Based on the extreme mutational sensitivity, high contact density of sidechain packing in the Kir2.2 crystal structure, and known disulfide-mediated fold stabilization, we propose that the top halves of M1 and M2, pore helix, and the M2-pore helix loop have critical roles in ion channel biogenesis and trafficking. This may be a general feature of K$^+$ channel folding as a similar 'cotranslationally folded biogenic unit' was proposed for Kv1.3, in which S5/pore loop/S6 interact to establish a 'reentrant pore architecture' and correct topology early in channel biogenesis without requiring tetramerization (*Delaney et al., 2014*; *Gajewski et al., 2011*). In KcsA, pore helix folding and tetramerization are uncoupled, with the selectivity filter being able to access nonnative, partially unstructured states during folding (*Song et al., 2021*). Consistent with these previous studies, residues from neighboring subunits that interact with this putative biogenic unit in the fully assembled channel are relatively tolerant to mutations (e.g., selectivity filter residues Y153, which forms a π–π interaction with F143 and F103). This supports the idea that early trafficking-critical folding events involve only intra-subunit interactions that promote tertiary structure formation within nascent Kir2.1 monomers (perhaps in the ER translocon) to establish correct topology and promote membrane insertion.

## Regions involved in gating transitions do not contribute to fold stability

From the perspective of surface trafficking as the measured phenotype, several regions of Kir2.1 are tolerant to mutations. This includes the interface between pore and CTD (comprised of slide helix αA/αB, inner helix gate [I183–M188], and tether helix αF), the G-loop (βH-βI loop), and the residues lining the CTD cavity (loops between βC-βD and βE-βG) at the interface between different subunits (*Figure 3E*). Interestingly, these are involved in gating-associated conformational changes (*Hansen et al., 2011*; *Li et al., 2015*) induced by the binding of PIP$_2$, Kir2.1's allosteric activator (two-sided Fisher's exact test for interrelation of mutational tolerance and gating region membership, p-value 8.091e-05, odds ratio 2.47). An explanation for high mutational tolerance in gating structural elements is that they have higher conformational flexibility required for interconversion between different gating states. They are therefore unlikely to contain motifs recognized by folding-based quality control mechanisms.

Taking the biogenic folding unit and the location of mutation tolerant regions together, Kir2.1 appears to be organized internally into contiguous regions involved in fold-stability (biogenic folding unit, IgG-like CTD) and gating transitions (TM/CTD interface, CTD subunit interfaces), each with distinct mutational tolerance. To understand the relationship between folding and function, we assayed K$^+$ conductance as a second phenotype for the DMS library.

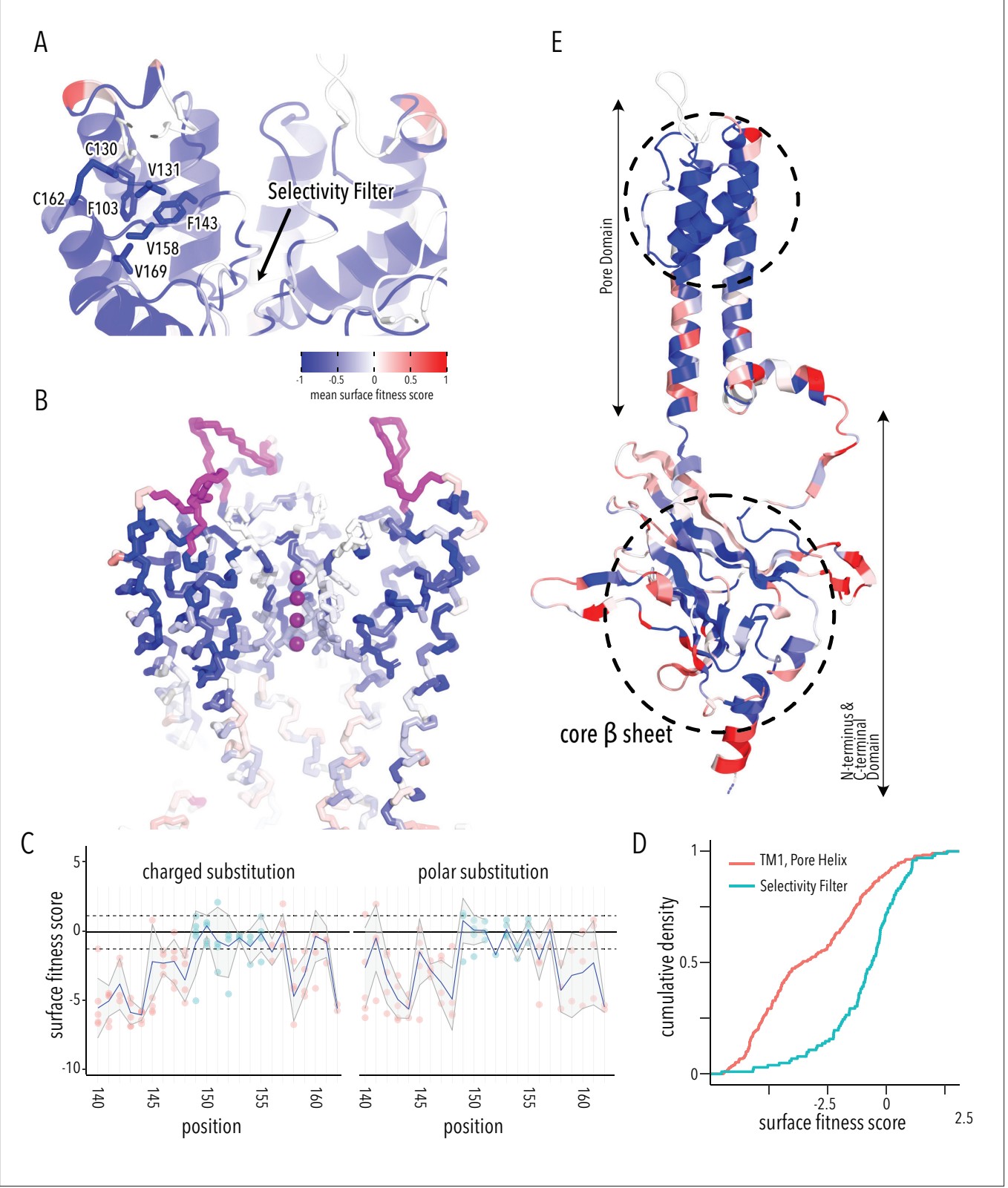

**Figure 3.** Determinants of improved surface expression and Kir2.1's biogenic folding unit. (**A**) Close-up of mean surface expression fitness scores in the pore domain. Residues that are part of the putative biogenic unit of Kir2.1 are shown as sticks. (**B**) Mean surface expression fitness of selectivity filter residues. K⁺ ions are shown as purple spheres. Regions with no data are shown in magenta. (**C**) Surface expression fitness of specific residues when replaced with charged (left panel) or polar residues (right panel). Selectivity filter residues are colored turquoise; neighboring residues are colored

*Figure 3 continued on next page*

*Figure 3 continued*

salmon. The solid horizontal line represents the median surface trafficking fitness of Kir2.1 synonymous mutations with dashed lines being ±1 SD. The gray-shaded ribbon represents a 90% confidence interval of mean surface fitness (n = 2–7) for each position (solid blue lines). (D) Cumulative density function of surface fitness score within the selectivity filter (turquoise line) and neighboring residues (salmon line). (E) Mean surface expression fitness scores mapped onto a single Kir2.1 monomer highlighting the importance of the biogenic folding unit in the pore domain and the core β-sheet of the C-terminal domain (CTD). All mapped mean surface fitness scores (averaged across all measured mutations for that position in **A**, **B**, and **E**) are represented as a gradient from blue over white to red, corresponding to below wildtype (WT), WT, and above WT fitness, respectively.

## Functional phenotype assays map molecular determinants of conduction and $PIP_2$ sensitization

To assay variant function, we sorted the Kir2.1 DMS library (in duplicate, on separate days) into populations based on voltage-sensitive dye fluorescence, sequenced, and calculated fitness using the same Enrich2 package. This assay is based on the premise that Kir2.1 decreases RMP, resulting in decreased fluorescence of the FLIPR dye. Functional fitness was determined for 6944 Kir2.1 variants (93.4%), with similar replicates (Pearson correlations 0.86–0.93, *Figure 4—figure supplement 1*) and excellent read depth (greater than 30-fold at most positions, *Figure 4—figure supplements 2–3*). Of the 475 missing variants (no data in either one or both replicates), only 158 were also missing in the stable library cell line, again suggesting stochastic dropout (*Figure 2—figure supplement 1*).

Function fitness median and standard deviation for synonymous and missense mutations were 0.07 ± 0.24 and 0.04 ± 0.6, respectively (*Figure 4A*, *Figure 4—figure supplement 4*). 10% percentile tail ends differed by 0.53 and 0.35 log units at the low and high ends, respectively, which suggest much less resolution compared to the surface trafficking assay. Unlike the bimodal distribution for surface expression fitness, functional fitness is unimodal, with most mutations being neutral (*Figure 4A*); there is no distinct low function population likely due to lower dynamic range of the voltage-sensitive dye compared to fluorescent antibody labeling (*Figure 1—figure supplements 3–4*). Function fitness is the combination of surface expression (number of channels on surface) and functional properties (open probability, single-channel conductance, $K^+$ selectivity, etc.). Furthermore, our assay is measuring steady-state RMP and even partially functional Kir2.1in the balance with onductances from other channels constiutively expressed within HEK293T cells could hyperpolarize cells. In aggregate, this means we must interpret functional fitness in the context of surface expression fitness.

Many variants in the pore domain have low functional fitness scores (e.g., pore helix, selectivity filter, *Figure 4B*). For example, Y153 mutations, which have relatively tolerant surface fitness, strongly reduce function (*Figure 4C*). Y153 interacts through π–π interactions with F143 and F103 on a neighboring subunit and mutations may preclude correct selectivity filter geometry to conduct $K^+$ ions. The conservative substitution Y153F has a neutral phenotype, in agreement with earlier electrophysiology studies (*Heginbotham et al., 1994*; *Splitt et al., 2000*). Many positions along the pore helix that form a hydrogen bond network to support a conductive and $K^+$ selective state (*Cheng et al., 2011*) are very sensitive to mutation. Mutations of conserved hydrophobic residues in the CTD core β-sheet (e.g., W220) and the Golgi export signal (e.g., hydrophobic cleft residues Y323, W330, and salt bridge R44, E327) are deleterious as predicted by their effect on surface expression.

Disordered N and C termini including the ER export signal FCYENE had moderately low surface expression fitness, but their functional fitness is mostly near-neutral (*Figure 4D and E*). This suggests that mutations within the disordered regions only impact trafficking of otherwise properly folded and functional channels. Mutations within the unstructured termini are deleterious to surface expression but neutral to channel function.

A specific structural mechanism for $PIP_2$ regulation was previously proposed (*Hansen et al., 2011*; *Tao et al., 2009*). $PIP_2$ binding at the TM/CTD interface induces a disorder-to-order transition in the tether helix, contracting it and thereby translating the CTD toward the pore domain. The engagement between CTD and pore domain allows the G-loop to wedge into the pore domain and the inner helix activation gate to open. As described by *Hansen et al., 2011*, $PIP_2$'s binding site includes pore domain residues (R80, W81, R82, K190) and CTD residues (K193, K195, K196, R67, R226, R197). We found that in these residues surface expression fitness was neutral and functional fitness was negative (*Figure 4F and G*). This gave us the idea that by focusing on mutation-sensitive positions (for function), we may be able to trace secondary structural elements that couple the $PIP_2$ binding site to the Kir2.1's gate, the G-loop (*Pegan et al., 2005*). Beginning with function-sensitive residues that directly

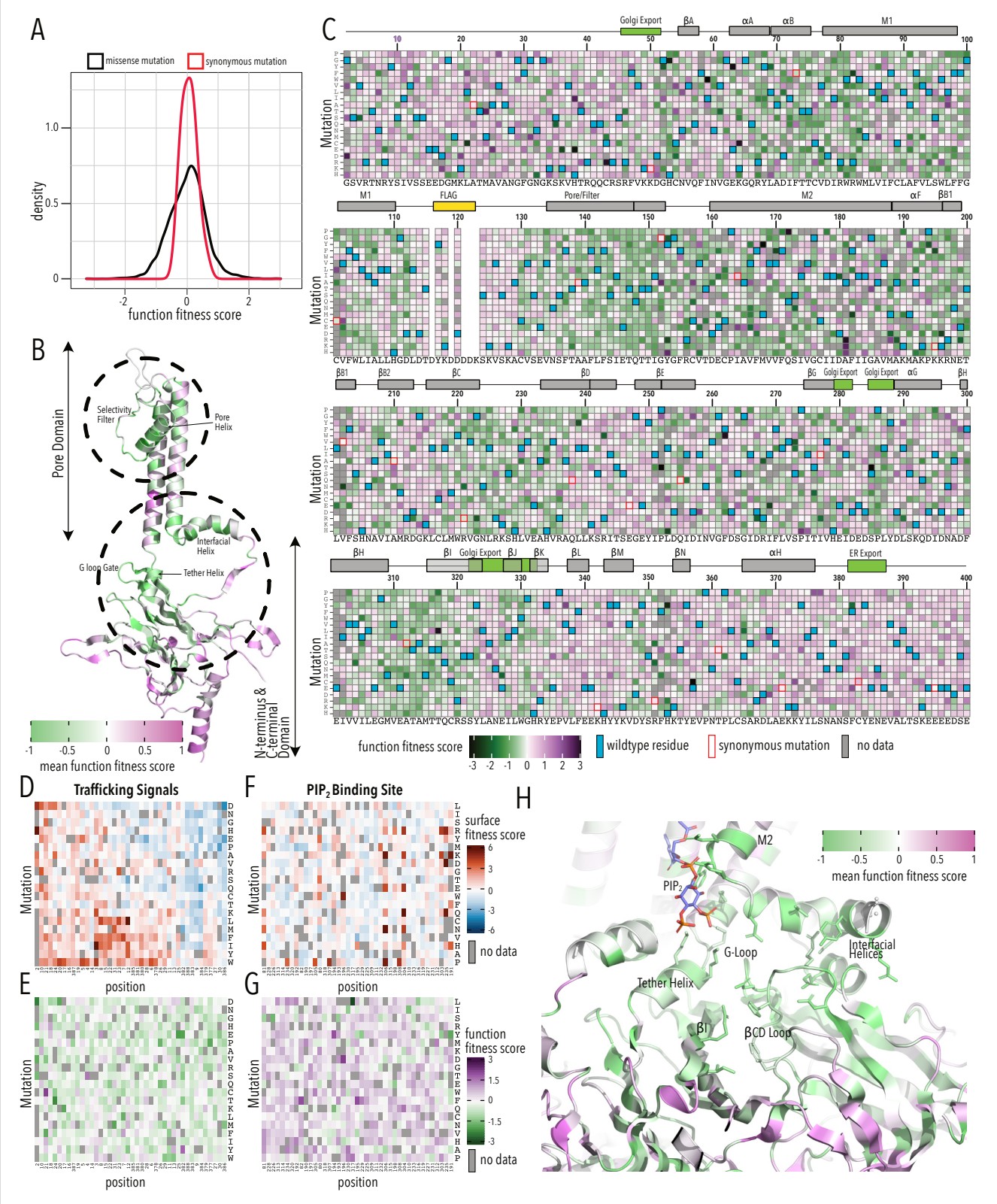

**Figure 4.** Function fitness. (**A**) Distribution of function expression fitness scores for synonymous mutations (red line) and missense mutations (black line). (**B**) Mean function fitness scores averaged across all measured mutations for that position mapped onto a single Kir2.1 monomer highlighting the importance of the pore helix, selectivity filter, and structural elements at the TM/C-terminal domain (CTD) interface (e.g., interfacial helix, tether helix, G-loop gate) for function. (**C**) Heatmap of function expression fitness scores. Wildtype (WT) residues are indicated by blue filled boxes. Missing data is

*Figure 4 continued on next page*

*Figure 4 continued*

indicated by filled dark gray boxes. Synonymous mutations are indicated by red box outlines. (**D–G**) Heatmap of fitness score in regions that contain bona fide trafficking signal (**D, E**; for surface and function scores, respectively) or the PIP$_2$ binding site (**F, G**; for surface and function scores, respectively). (**H**) Close-up of function fitness at the TM/CTD interface. Allosteric ligand PIP$_2$ is shown as sticks. Surface fitness in (**D, F**) is represented as a gradient from dark blue over white to dark red, while function fitness in **B, C, E–H** is represented as a gradient from green over white to magenta. Both gradient colors order corresponds to below WT, same as WT, and above WT fitness, respectively.

The online version of this article includes the following figure supplement(s) for figure 4:

**Figure supplement 1.** Function fitness assay biological replicates.

**Figure supplement 2.** Function fitness assay biological replicates.

**Figure supplement 3.** Function assay read statistics and assay repeatability.

**Figure supplement 4.** Function fitness assay standard error distribution.

---

interact with PIP$_2$ in the pore domain (R80, W81, R82) and tether helix (K190, K193, K195, K196), we find additional residues in the tether helix (R197) that are in close vicinity to function-sensitive residues in neighboring structural elements (L201 in βB1, R67 in the interfacial helix). Both βB1 and interfacial helix form structural contacts with the βCD loop (*Hansen et al., 2011*; *Whorton and MacKinnon, 2011*), and prior functional studies have established the role of βCD in mediating PIP$_2$ regulation (*D'Avanzo et al., 2013*; *Lopes et al., 2002*). In agreement with these studies, residues are highly function-sensitive in the entire βCD loop (*Figure 4H*). Function sensitivity extends to residues on the βCD loop (e.g., H229) and βEG loop (e.g., F270) of one subunit that potentially couple to the βH (I305), βI (R320), and the G-loop gate of another subunit, thus completing the trace to Kir2.1's gate.

Taken together, DMS aligns with established mechanistic knowledge and structural connectivity, supporting the central role of the βCD loop propagating PIP$_2$ binding at the TM/CTD interface to the G-loop gate. Molecular dynamics simulations uncovered an extended network of interactions between the N-terminus, tether helix, and CD loop to keep the G-loop gate in a closed state (*Li et al., 2015*). Electrophysiology demonstrated that mutations in R218 (R226 in Kir2.1-Flag numbering), which is conserved in the K$_{IR}$ family, attenuate PIP$_2$ affinity (*Lopes et al., 2002*) to cause Anderson–Tawil syndrome (*Plaster et al., 2001*). Mutating the corresponding residues in Kir3.2 – R201A – constitutively activates channels, potentially by forcing the βCD loop into a conformation that mimics the G-protein-activated state (*Whorton and MacKinnon, 2011*).

Another striking feature revealed by functional assays are function-increasing mutations near the helix bundle crossing gate toward the lower half of TM2 (*Figure 5A and B*). Previous studies suggested that this 'lower glycine hinge' is required for tight packing of M2 helices (*Shang and Tucker, 2008*). Mutation to aspartate (G178D in Kir2.2) resulted in a 'forced-open' channel (*Zangerl-Plessl et al., 2020*). In agreement, G185D (the corresponding residue in Kir2.1-Flag, and – more strongly – G185T) increased function fitness scores (*Figure 4C*). Several other, but not all, mutations above and below G185 also increase functional fitness (e.g., I184H, M188R, M191Y). Together, these residues form a cuff above the HBC gate coupling TM2 from different subunits (*Delaney et al., 2014*; *Doyle et al., 1998*). The cuff may stabilize specific conformations of M2 that decrease open probability by keeping the HBC gate closed; mutations to large residues may disrupt this hydrophobic interaction network with a function-increasing effect. Additional electrophysiological characterization is needed to independently test this idea. The HBC gate is also stabilized by hydrophobic contacts to M1 of an adjacent subunit (*Zangerl-Plessl et al., 2020*; *Meng et al., 2016*). Consistent with M1 interactions forming a gating energy barrier, mutating interacting TM1 residues (F88) also increases function fitness scores.

To identify regions that are important for function and not surface expression, we compared surface fitness and function fitness for each position (*Figure 5C*). For many positions, including the FLAG-tag, N-termini, and Golgi export motifs, surface scores were strongly negative (deleterious) while function scores were relatively neutral (*Figure 5C*, positions with orange lines). This apparent discordance is likely due to the difference in measurement dynamic range between surface expression and function assay. The function assay that is measuring steady-state RMP and fully functional Kir2.1, even at lower surface expression level, may hyperpolarize the cell, giving the appearance of a neutral phenotype. Transmembrane helices and CTD core β-sheets had low surface score and low function scores (*Figure 5C*, positions with blue lines). For another subset of positions, function scores were

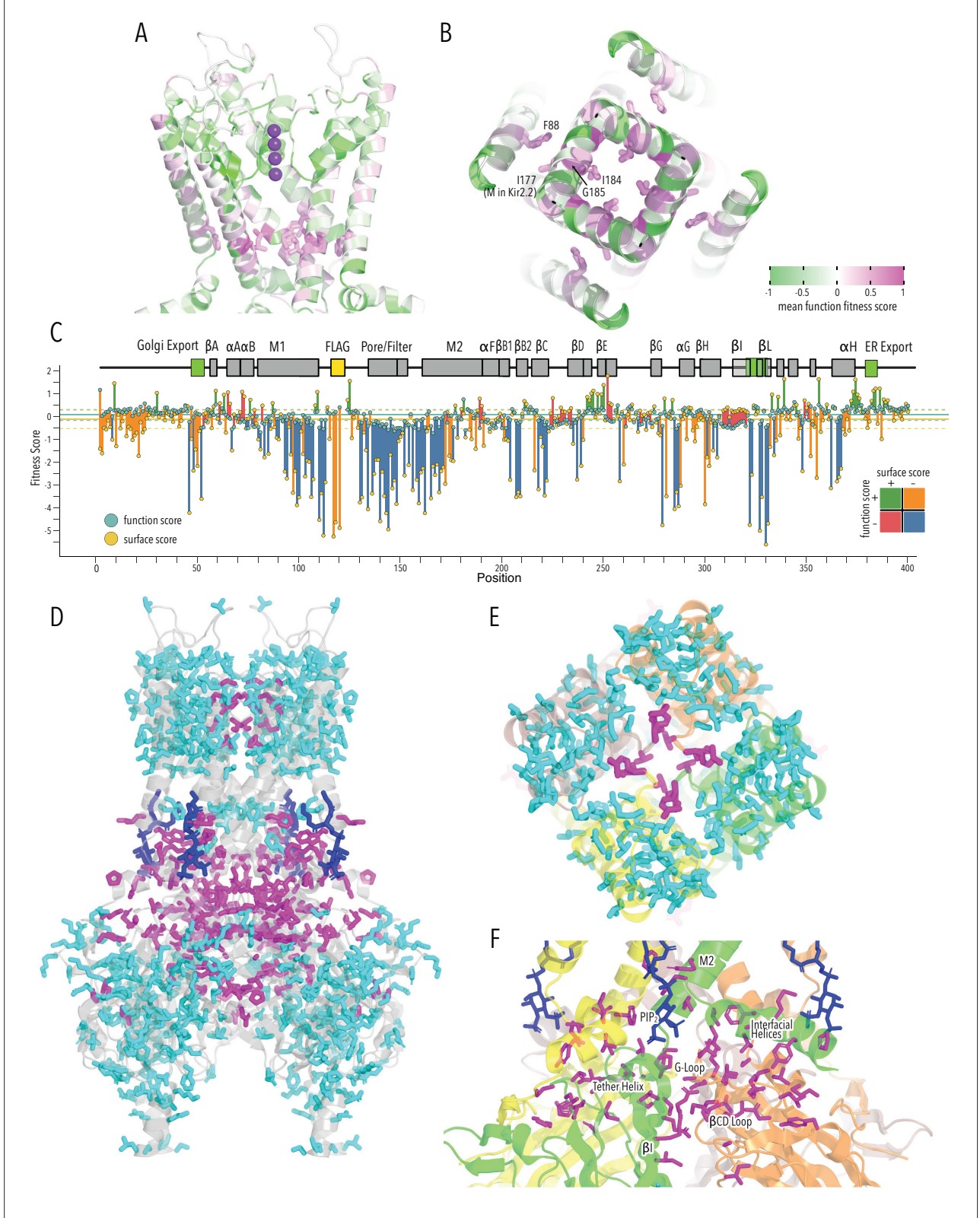

**Figure 5.** HBC gate mutations and regions with distinct roles in stability and function. (**A**) Function fitness (averaged across all measured mutations for that position) in the pore domain represented as a gradient from green over white to magenta, corresponding to below wildtype (WT), same as WT, and above WT fitness, respectively. One subunit is removed for clarity. K⁺ ions in the selectivity filter are shown as purple spheres. Sidechains of residues that are enriched for activating mutations are shown as sticks. (**B**) Detailed view of the HBC gate. Numbering corresponds to Kir2.1-Flag. (**C**) Surface fitness

*Figure 5 continued on next page*

*Figure 5 continued*

(orange circles) and function fitness (teal circles) for each position. The line length between each score represents the score difference and line color represents sign combination of both scores (both negative; one negative, one positive; etc.). Vertical solid and dashed lines represent median scores ±1 SD of synonymous Kir2.1 mutations for surface fitness (orange) and function fitness (teal). Kir2.1 secondary structural element are shown as gray boxes; trafficking signals are colored green. (D–F) Residues with significant function/surface fitness score differences are mapped onto the Kir2.2 structure (PBD 3SPI). Residues with mutations that predominantly reduced function are shown in magenta. Residues with mutations that predominantly reduced surface expression are shown in cyan. PIP$_2$ is shown in blue. Overview (**D**), close-ups of selectivity filter (**E**), and TM/C-terminal domain (CTD) (**F**).

The online version of this article includes the following figure supplement(s) for figure 5:

**Figure supplement 1.** Comparison of intra- vs. intersubunit contact fitness scores.

significantly lower than surface expression scores (*Figure 5C*, positions with red lines). This includes sites involved in gating (PIP$_2$ binding, G-loop gate). By making a binary assignment whether a position belongs is 'surface trafficking and function-sensitive' or 'function-sensitive' and mapping this assignment onto the Kir2.2 structure, we again find structural organization around elements that are important for fold stability (TM, CTD β-sheets) and function (PIP$_2$ sensitization, gating, K$^+$ conduction) (*Figure 5D*).

This distinct organization is clearest in the pore domain, where all interactions within a subunit are driving surface expression scores (*Figure 5E*, cyan sticks), while residues mediating subunit interactions required for the selectivity filter stability are driving function scores (*Figure 5E*, magenta sticks). Other residues that drive function scores are distributed along an interaction network that connects the PIP$_2$ binding site (M2, αF) with the slide helix (αA/B), βCD loop, βEG loop, and the G-loop gate (βHI loop) (*Figure 5F*). Intriguingly, this network involves interfaces between adjacent subunits (N-terminus<>βLM loop interface, βI<>βCD loop, βD<>βEG loop). Subunit interactions determining gating is consistent with recent crystal structures of a forced-opened Kir2.2 in which inner helix gate opening requires CTD subunits to move relative to each other (*Zangerl-Plessl et al., 2020*).

For a more quantitative comparison, we used the Protein Contact Atlas (*Kayikci et al., 2018*) to systematically identify noncovalent intra- and intersubunit contacts in Kir2.2 (using both closed-state PDB 3SPI and forced-open state PDB 6M84; *Figure 5—figure supplement 1*, *Supplementary file 2*). Taking only residues in the CTD into account, we find that residues that participate exclusively in intra-subunit contacts are drivers of surface expression fitness. With surface fitness scores as the metric, residues that make inter-subunit contacts are much more tolerant to mutations (*Figure 5—figure supplement 1D*; two-sample Kolmogorov–Smirnov test p-value<2.2e-16). Conversely, function scores of mutations in residues that do make inter-subunit contacts are significantly lower than those that make only intra-subunit contact (*Figure 5—figure supplement 1E*, two-sample Kolmogorov–Smirnov test p-value 3.713e-10). However, this effect is less severe than for surface fitness, probably because a properly folded state is a prerequisite for proper function. Residues that exclusively make inter-subunit contacts or make no contacts at all are enriched for neutral/moderately function improving mutations.

Overall, the function assay allows us to validate and test existing models of Kir2.1 gating while also providing further evidence that inter-subunit interactions bias Kir2.1's conformational ensemble toward closed states.

## Surface and Function DMS results agree with prior studies

To provide a more critical assessment of our surface and function fitness dataset quality, we compared our per-variant scores with equivalent data from prior studies. Ma et al. previously had performed an alanine scan in Golgi export motifs and measured cell surface expression (*Ma et al., 2011*). Their fraction of surface expression compared to WT and our surface fitness score are strongly positively correlated (Pearson correlation coefficient 0.72, *Figure 6A*). Dart et al. had previously used scanning cysteine mutagenesis of the selectivity filter and Ag$^+$-blockage to map which residue lines the ion conduction pathway (*Dart et al., 1998*). We find that Ag$^+$-mediated inhibition is strongly positively correlated with surface expression fitness (Pearson correlation coefficient 0.75, *Figure 6B*.) This is consistent with the idea that buried residues – those involved in structural stability of the pore domain – are drivers of surface fitness score. Of course, we expect a negative correlation if we compare Ag$^+$ block to function scores. However, we need to account for proper folding being a prerequisite for function. We therefore compared the ratio of function/surface fitness score with Ag$^+$ block and found strong negative correlation. Corrected for the effect on surface fitness, residues important for

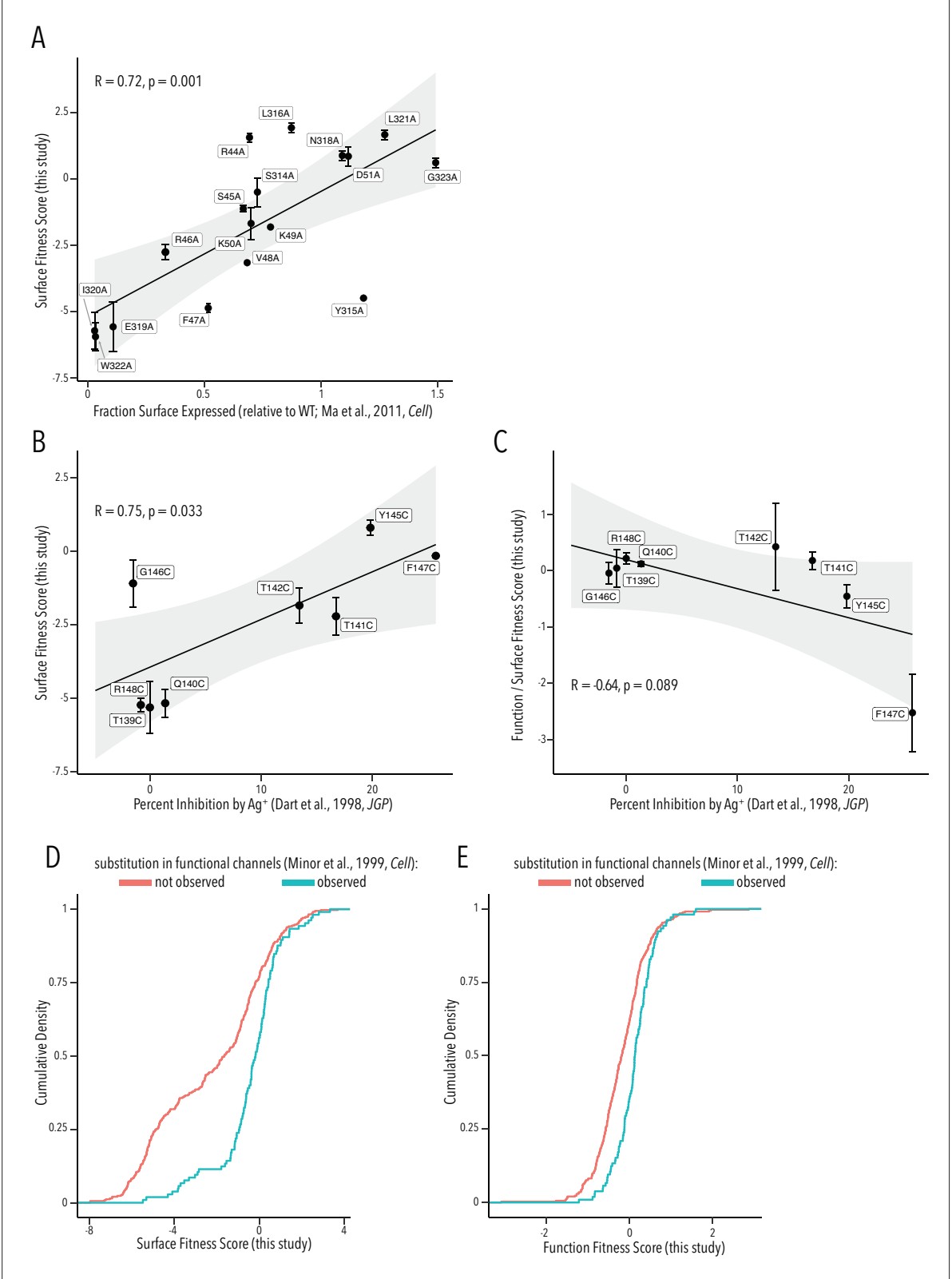

**Figure 6.** Comparison to prior studies. (**A**) Scatterplot of surface-expressed fraction (compared to Kir2.1) for alanine mutants tested in *Ma et al., 2011* vs. their corresponding surface fitness score measured in this study. (**B, C**) Scatterplot of Ag$^+$-mediated Kir2.1 inhibition report in *Dart et al., 1998* vs. their corresponding surface fitness score (**B**) or surface fitness-normalized function score (**C**) measured in this study. For (**A–C**), the solid black line represents a linear regression with the gray-shaded ribbon being the 95% confidence interval. Pearson correlation coefficient and p-value are shown in

*Figure 6 continued on next page*

Figure 6 continued

inset. Numbering corresponds to Kir2.1-Flag. (**D, E**) Cumulative density function of surface scores (**D**) or function scores (**E**) for observed substitution (teal line) or not observed substitutions (red line) in *Minor et al., 1999*. All error bars represent Enrich2-calculated weighted-least-squares regression standard error.

function (higher Ag$^+$ block) have lower function scores in our assay (**Figure 6C**). In a third comparison, we turned to *Minor et al., 1999*, who previously had used random mutagenesis of Kir2.1's transmembrane domains. By using Sanger sequencing on colonies grown expressing Kir2.1 variants in a K$^+$ transport-deficient yeast strain under selective conditions, they recovered frequency counts for substitutions compatible with forming functional channels. By comparing surface and function fitness to Minor et al.'s observed vs. nonobserved substitutions (presumably meaning unallowed mutations), we find that observed substitutions are enriched for more neutral fitness scores (two-sample Kolmogorov–Smirnov test, p-value 7.65e-11 and 3.584e-07, respectively) (**Figure 6D and E**).

Taken together, we find good agreement between prior studies and our surface fitness and function fitness scores. This suggests our datasets provide sufficient sensitivity, dynamic range, and fidelity for meaningful interpretation and hypothesis formation.

## Most pathogenic inward rectifier mutations are clustered within functionally important residues

A central goal in molecular genetics is identifying the mechanistic basis by which mutations cause disease. Multiparametric DMS studies that interrogate multiple phenotypes associated with variants are a promising strategy to answer this question. Based on the premise that most pathogenic mutations affect protein stability, approaches such as Vamp-Seq (*Matreyek et al., 2018*) were developed as generalizable measures of protein abundance. In cases such as Kir2.1, where proper trafficking, localization, and gating are crucial for producing functioning proteins, measuring abundance likely will not work. It is necessary to differentiate between mutations' effects on folding, trafficking, and gating to learn the mechanism for how genotype affects Kir2.1 phenotype. To learn potential pathogenicity and mechanism of action for reported disease-associated mutations, we compared the surface expression and function fitness data to clinically observed mutations.

Since the overall domain architecture and structure/function relationships are conserved within the inward rectifier family, we reasoned that the mechanisms underlying variant effects are conserved, as well. We therefore gathered missense variant effects reported in ClinVar (*Landrum et al., 2018*) and gnomAD (*Karczewski et al., 2020*) for inward rectifiers (as of March 25, 2022). By aligning all human Kir, we assigned the corresponding Kir2.1 position to each variant and noted whether the wildtype amino acid matches between Kir2.1 and the aligned Kir (total variant count: 2613; *Supplementary file 1*). To test if variants in ClinVar or gnomAD are related to surface trafficking or function, we compared the trafficking and function scores variants to those in databases vs. those that are not. They differed for trafficking scores but not for function scores (two-sided Kolmogorov–Smirnov test p-values 0.005563 and 0.7298, respectively; *Figure 7A and B*). This means being listed in genetic variation databases is related to Kir2.1 function and less related to trafficking. Variants with low surface scores in our DMS are underrepresented in databases. A likely explanation is that Kir2.1 is essential for normal physiology and therefore under strong selection. Indeed, in support of Kir2.1's essential role, homozygous knockout mice die 8–12 hr after birth (*Zaritsky et al., 2001*) and Kir2.1 has a low missense constraint score of 0.5 within gnomAD (*Karczewski et al., 2020*). Variants that misregulate surface trafficking are more likely more deleterious (e.g., abolishing all K$^+$ conductance) than variants affecting function (e.g., gating kinetics), which is why the former are extremely rare in the living human population. Consistent with this idea, exome and targeted sequencing in products of conception showed a significant enrichment of pathogenic variants associated with cardiac channelopathies in stillbirths (*Sahlin et al., 2019*). Furthermore, only two variants of the five most surface-trafficking impairing positions on Kir2.1's Golgi export motif (R46, S322, E327, I328, W330, G331) are reported in ClinVar (S322F uncertain significance, W330C pathogenic). Another variant (R46H) is only observed as heterozygous in gnomAD.

The correspondence between variant listing and inward rectifier function is also apparent when we annotate mean surface trafficking and function scores with pathogenic variants (*Figure 7C*). Across the board, hotspots enriched for pathogenic disease-associated mutation have low function scores,

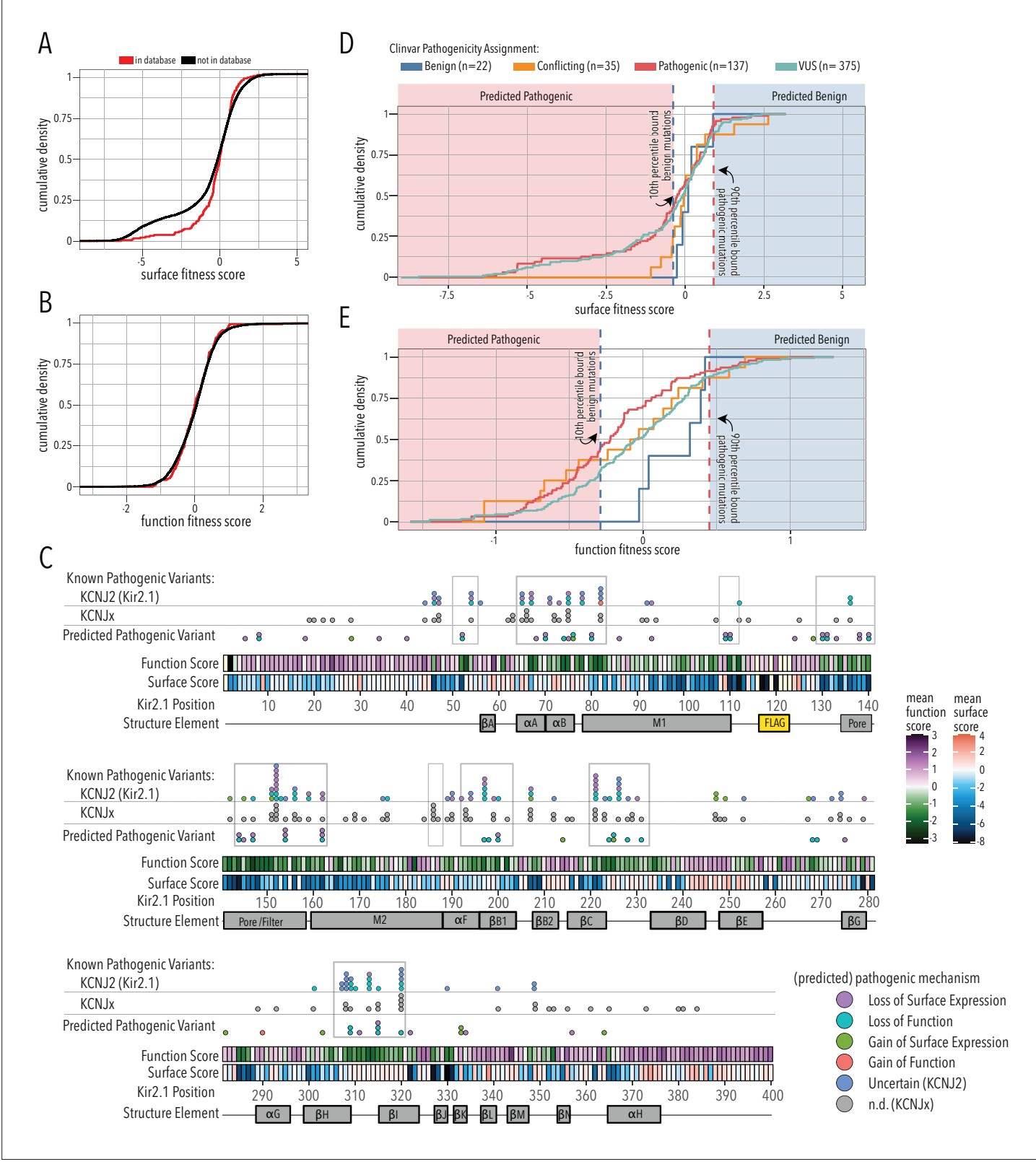

**Figure 7.** Folding and trafficking variants in clinical databases. Cumulative distributions of surface (**A**) or function (**B**) fitness scores for variants that are represented in ClinVar and gnomAD (red line) or that are not (black line). (**C**) Line plot by position of mean function or surface fitness scores. Fitness scores below wildtype (WT), same as WT, and above WT fitness are represented as a gradient from dark blue over white to dark red for surface fitness and green over white to magenta for function fitness, respectively. Kir2.1 secondary structural elements are shown as gray boxes. In the top row, known

*Figure 7 continued on next page*

*Figure 7 continued*

pathogenic variants in Kir2.1 (*KCNJ2*) are mapped as circles filled with color indicating predicted pathogenic mechanism determined in this study. In the row below, known pathogenic mutations in other Kir (*KCNJx*) are mapped as gray circles. In the third row, reported variant of unknown significance (VUS) are mapped as circles filled with color indicating predicted pathogenic mechanism determined in this study. Mutation hotspots are indicated by gray outlines. (**D, E**) Distribution of surface fitness (**D**) and function fitness (**E**) by ClinVar pathogenicity assignment (represented as line color; 'Conflicting' means some studies interpreted a variant as benign, others as pathogenic). Vertical dashed lines represent 10th and 90th percentile boundaries of expert reviewed benign variants (blue) and pathogenic variants (blue), respectively. Variants with score below the 10th percentiles of known benign variant bound are predicted to be pathogenic (shaded red zone), while variants above the 90th percentile of known pathogenic variants are predicted to be benign (shaded blue zone).

The online version of this article includes the following figure supplement(s) for figure 7:

**Figure supplement 1.** Pathogenic variant frequency tables.

whereas their surface expression scores are more varied. Variants of uncertain significance followed this trend. In addition to regions involved in PIP$_2$ binding (αA/B slide helix, lower part of M1, αF), and propagating conformational changes to the G-loop gate (βB1, βCD, βHI loop), pore helix and selectivity filters are also hotspots for pathogenic variants.

We further divided fitness scores by assigned clinical significance to estimate the fitness score bounds of expert-reviewed benign and pathogenic variants (***Figure 7D and E***). For surface fitness, we find that >90% of known benign variants have a fitness score >–0.3. With this lower bound assumption for benign variants, we predict that any 'variant of unknown significance' (VUS) scoring lower will be pathogenic with a loss of surface expression phenotype. Conversely, the upper surface fitness score bound (10th percentile) for expert-reviewed benign variants was 0.92, and VUS with greater surface scores are predicted to have a gain of surface expression pathogenic phenotype. We applied the same reasoning to function fitness scores. Using this estimate, we can assign a predicted pathogenic phenotype (loss/gain of surface expression/function) to 75 of 89 (76%) Kir2.1 VUS and variants with conflicting interpretation; several have multiple predicted phenotypes (***Supplementary file 3***). For example, the M1 mutation R80C or the G-loop mutation M307I, which are associated with short QT syndrome 3 and long QT syndrome, respectively, have both predicted loss of function and loss of surface expression phenotypes. In another example, C76C, located in the slide helix, has a gain of surface expression but loss of function predicted phenotype. The primary sequence location of these predicted pathogenic variants is shown in ***Figure 7C***. Many predicted loss of function variants localize to the slide and tether helices, G-loop gate, and selectivity filter/pore helix. Gain of surface expression variants map to the βBC and βK loop near the interface to βDE of a neighboring subunit. The putative biogenic folding unit (top of M1, M2, and pore helix) contains several predicted loss of surface expression variants. Interestingly, no expert-reviewed pathogenic variants map to this region (apart from selectivity filter G152 and Y153), which suggests that DMS-based prediction can identify new mutation hotspots. We can also use estimated bounds of benign variants to predict the mechanism of 26 of the 53 expert-reviewed pathogenic or conflicting variants that cause Kir2.1 disorders (***Supplementary file 4***). For six of these mutations, a combination of both loss-of-function plus loss of surface expression is predicted.

When we compare the number of predicted phenotypes, we find that the odds are much higher to predict a loss of surface expression vs. loss of function in experimentally measured variants compared to those reported in ClinVar (one-sided Fisher's exact test, p-value 0.0001145, odds ratio 4.6, ***Figure 7—figure supplement 1***). This again suggests that loss of surface expression variants are underrepresented in ClinVar.

Taken together, we find that variants with deleterious surface score are underreported in available databases, which we propose is due to the essential nature of Kir2.1 in human physiology. Using bounds of surface and function scores for expert-assigned variants, we can predict VUS effects. We can also predict the mechanism of action through which known pathogenic mutations cause Kir2.1 disorders. This suggests that DMS-derived surface trafficking and function scores can be useful to predict pathogenicity and underlying mechanism of inherited and de novo mutations. Because of the conserved architecture of the inward rectifier family, assignment of mutation hotspots may extend to the entire inward rectifier family. Further corresponding DMS studies in other inward rectifiers are required to separate generalizable themes from idiosyncrasies. For example, Kir6.2 requires binding to the ABC transporter SUR1 to be expressed to the cell surface and function. This means

that assay-reported phenotypes for a subset of variation at the Kir6.2/SUR1 interface (involving αA/B slide helix, M1, βLM sheets) have greater bearing in Kir6.2-linked diseases (e.g., diabetes mellitus). The methods described here are a blueprint for these studies.

## Discussion

DMS quantitatively links protein phenotypes to the genetic variation within the protein's coding sequence. This style of systematic amino acid substitution can reveal protein function (*Araya et al., 2012*; *Fowler et al., 2010*; *Starr et al., 2020*), determine protein structure (*Schmiedel and Lehner, 2019*), and explain protein behavior in healthy and diseased cellular contexts (*Matreyek et al., 2021*). The phenotypic outcome of genetic variation can be multifaceted. Folding, membrane trafficking, and functional modification of a protein can occur through multiple mechanisms such as protein–protein interactions (e.g., chaperones) or ligand binding (e.g., allosteric modulators). To explore the numerous impacts of mutations, we measure two phenotypes of Kir2.1 variation: surface trafficking and $K^+$ conduction. Both assays are based on cell sorting mediated by fluorescent signals (antibody binding to an extracellular loop and a voltage sensor dye). By measuring how variants are enriched or depleted, we assign quantitative fitness scores.

Many other assay types are compatible with this approach, providing opportunities for even richer phenotypic description of ion channel variation. For example, a recently described approach, HiLITR (*Coukos et al., 2021*), could provide more granular resolution about trafficking motifs and Kir2.1 localization in living cells. Spontaneously-spiking HEK cells (*Hochbaum et al., 2014*; *Park et al., 2013*), which co-express Kir2.1, a voltage-dependent $Na^+$ channel (Nav1.7), the channelrhodopsin CheRiff, and genetically encoded voltage indicator QuasAr2, could be adapted to evaluate how ion channel expression levels and gating properties impact excitability (*O'Leary et al., 2014*). When assays are conducted in combination with small molecule screening, they may aid in the discovery of novel allosteric modulators, state-dependent blockers, and molecular chaperones to precisely treat channelopathies based on genotype. Integration of disparate assays into a common framework of protein variation and its role in structure and function will be challenging, but efforts to standardize reporting and unified statistical frameworks for interpretation (e.g., Enrich2 [*Rubin et al., 2017*], the Atlas of Variant Effects [*Fowler et al., 2021*]) are on the horizon.

While our assays were done in an immortalized nonpolarized HEK293 cell line, which is a caveat in trafficking and function fitness, many of our findings with respect to mutational sensitivity of trafficking motifs, core structural elements, and regions of Kir2.1 involved in gating align with existing knowledge and biophysical intuition. Unlike previous more limited screens and intuition, our data is quantitative and enables data-driven approaches to construct global biophysical models of Kir2.1 structure and function in this specific cellular context. For example, the Golgi export signal comprised of two patches at the N-terminus and CTD was first described as a minimal set of hydrophobic residues along a CTD cleft and juxtaposed basic residues (*Ma et al., 2011*) and later expanded to adjacent sites (*Li et al., 2016*). Our comprehensive screen shows that these trafficking motifs may indirectly probe the correct folding of the entire β-sheet core of the CTD. This motif appears integral to allowing Kir2.1 to pass intracellular 'quality control' after having achieved proper conformation of the entire CTD and assembly into tetramers.

We observed contiguous regions with neutral surface expression fitness involved in gating transitions while regions involved in fold stability were highly sensitive to mutations. Looking at the same regions in functional data, we find the inverse. This suggests two things. First, comprehensive assessment of Kir2.1's phenotypes after perturbation (i.e., mutation) is a high-throughput method to annotate protein sequences into classes linked to specific functions (e.g., putative trafficking signals, folding units, etc.). Conceptually this is similar to other high-throughput biochemical approaches that probe sequence–function relationships, such as circular permutation profiling (*Atkinson et al., 2018*) or high-throughput enzyme variant kinetics measurements (*Markin et al., 2021*). Second, combining multiple assayed parameters is key to discover general organizational principles in Kir2.1, specifically two distinct structural regions with distinct roles in providing fold stability or dynamics required for gating transitions. Expanding to more measured phenotypes, in different cellular contexts, and integrating datasets may be the blueprint for 'sequencing-based' biophysics that probes protein function, folding, and dynamics through steady-state biochemical experiments.

Our DMS study provides additional context for the mechanistic basis of structure/function relation (e.g., reaffirming the βCD loop's role in propagating PIP$_2$ binding at the TM/CTD interface to the G-loop gate). Our data is consistent with a biogenic folding unit that represents an early quality control step of reentrant pore architecture and correct topology, likely while the monomer is in the translocon. A similar biogenic unit was described in voltage-dependent K$^+$ channels, where it is stabilized by an extensive network of interactions (*Delaney et al., 2014*). Furthermore, many ion channels have structurally homologous reentrant pore loop architectures and earlier studies suggested that the presence of an 'aromatic cuff' is a general feature of their biogenesis (*Delaney et al., 2014*; *Doyle et al., 1998*). With further assay development to separate variant effect on folding vs. export and trafficking between different cell organelles, DMS studies may provide a path to probe the energetics and biophysics of stabilizing interactions and to test the hypothesis that hydrophobicity is a general stabilizing factor across reentrant pore loop architectures.

DMS may also shed light on how subunit interactions determine inward rectifier gating properties. We find that mutations near the interface of subunits strongly increase Kir2.1 function. This is consistent with subunits' roles in setting channel gating properties. Single-channel patch electrophysiology of Kir2.1/2.2 heterotetramers showed that addition of a Kir2.2 monomer increases single-channel conductance and decreases the open dwell time ($\tau_{open}$) (*Panama et al., 2010*). Perhaps this feature contributes to the observed differences in gating between inward rectifier homo- and heterotetramer, which differ in number and nature of subunit interactions.

Our finding that variants with deleterious surface score are underreported in available database supports the emerging theme that many disease-causing variants are linked to trafficking defects (*O'Donnell et al., 2017*; *Peters et al., 2003*; *Fallen et al., 2009*; *Zangerl-Plessl et al., 2019*; *Li et al., 2016*; *Ma et al., 2001*; *Ma et al., 2002*; *Ma et al., 2011*; *Stockklausner et al., 2001*; *Zerangue et al., 1999*; *Lin et al., 2006*). Other large-scale mutational analysis, as undertaken for the voltage-dependent K$^+$ channel Kv11.1 (*Anderson et al., 2014*; *Kozek et al., 2020*), has similarly shown that 88% of long QT-linked variants have trafficking-deficient mechanisms. They also demonstrate that data-driven approaches outperform smaller, more limited studies that predicted normal trafficking for most mutants (*Harley et al., 2012*).

Underrepresentation in variant databases has implications for clinical practice since standards for the interpretation of sequence variants are heavily focused on null variants (nonsense, frameshift), population frequency (frequent variants are likely benign), predicted functional effect (is the variant in an important domain), and case evidence (*Richards et al., 2015*). In genes under strong purifying selection, such as *KCNJ2*/Kir2.1, missense mutations cause severe developmental defects (craniofacial structures, limb development; *Zaritsky et al., 2001*; *Belus et al., 2018*) that likely contribute to or cause spontaneous abortions and pregnancy loss. These variants therefore never enter population or clinic-associated variant databases. DMS studies are not limited by organismal viability to probe the structural and functional consequences of genetic variation. DMS could be useful to identify drivers of spontaneous abortions and the underlying mechanisms of recurrent pregnancy loss.

By comparing our experimentally determined fitness scores to expert-reviewed assignment of variants effects, which are often based on smaller-scale direct biochemical studies, we can estimate the bounds of fitness scores for benign and pathogenic variants. This allows us to make predictions about pathogenicity and mechanisms of action for VUS and known pathogenic variants, closing the loop between high-throughput assays, biophysical mechanisms underlying fitness scores, and clinical interpretation of human variation in ion channel genes.

## Materials and methods
### Kir2.1 DMS library generation

Into mouse Kir2.1 (UniProt P35561), we introduced a FLAG-tag into an extracellular loop (at position T115) and added a downstream expression marker miRFP670 co-expressed via a P2A sequence. For Kir2.1 residues 2–391, each wildtype amino acid was mutated to all other 19 amino acids weighted by their codon usage frequency in humans. In addition to these missense mutations, mutations synonymous to wildtype were included for 20 positions as a benchmark. We included missense mutation into the FLAG as a negative control. We generated this mouse Kir2.1 DMS library using the SPINE (*Coyote-Maestas et al., 2020*; *Nedrud et al., 2021*), which we briefly

summarize as follows: SPINE-based libraries employ synthesized pools of DNA oligos with mutations at each position of a gene. Due to high error rates in oligo synthesis, the current maximum length for oligos from Agilent is 230 base pairs. As most genes are longer than 230 base pairs, we break up our gene (eight sections in the case of Kir2.1) and replace a subsection of the gene with a pool of mutated oligos. The oligos are designed with unique barcodes for amplifying out a specific subpools library, Golden Gate-compatible BsmBI cut sites, and the mutation within a subsection of Kir2.1. OLS oligos were designed using the SPINE scripts on GitHub (https://github.com/schmidt-lab/spine, *Coyote-Maestas and Nedrud, 2022*). These scripts design oligo libraries, primers for amplifying oligo-sublibraries, and inverse PCR primers for adding compatible cut sites to the Kir2.1 plasmid.

All backbones were amplified using a 25-cycle PCR with GXL polymerase and 1 ng of backbone DNA as template. The PCR product was then gel-purified. All oligo libraries were amplified using 25-cycle PCR with GXL polymerase and with 1 µl of the OLS library (resuspended in 1 ml TE) as template. To assemble the backbone and library DNA, Golden Gate reactions were set up in 20 µl containing 100 ng of amplified backbone DNA, 20 ng of amplified oligo DNA, 0.2 µl BsmBI -HFv2 (New England Biolabs), 0.4 µl T4 DNA ligase (New England Biolabs), 2 µl T4 DNA ligase buffer, and 2 µl 10 mg/ml BSA. These reactions were put in a thermocycler overnight using the following program: (1) 5 min at 42°C, (2) 10 min at 16°C, (3) repeat 40 times, (3) 42°C for 20 min, and (4) 80°C for 10 min. This reaction was cleaned using a Zymo Research Clean and Concentrate 5 kit and eluted in 6 µl of elution buffer. The entirety of this reaction was transformed in E. cloni 10G electrocompetent cells (Lucigen) according to the manufacturer's instructions. Cells were grown overnight with shaking at 30°C to avoid overgrowth in 30 ml of LB with 40 µg/ml kanamycin and library DNA was isolated using Zymo Zyppy miniprep kits. A small subset of transformed cells was plated with varying dilutions to assess transformation efficiency and validate successful mutations. All libraries at this step yielded >300,000 colonies implying a 100× coverage (assuming 1/3 of the variants were perfect based on our previous analysis of Agilent OLS-based libraries). Each sublibrary was combined at an equimolar ratio to make a complete library with all intended mutations included.

## Stable cell line generation

To generate cell lines, we used a rapid single-copy mammalian cell line generation pipeline (*Matreyek et al., 2020*). Briefly, mutational libraries are cloned into a staging plasmid with BxBI-compatible *attB* recombination sites using BsmBI Golden Gate cloning. We amplify the staging plasmid backbone using inverse PCR and the library of interest with primers that add complementary BsmBI cut sites. Golden Gate cloning and subsequent transformation was conducted with BsmbI (NEB), T4 Ligase (NEB) following the manufacturer's instructions using the same protocol as previously described for library generation. Completed library landing pad constructs are co-transfected (1:1) with a BxBI expression construct (pCAG-NLS-Bxb1) into (TetBxB1BFP-iCasp-Blast Clone 12 HEK293T cells) using Turbofect according to the manufacturer's instructions in six wells of a six-well dish. This cell line has a genetically integrated tetracycline induction cassette, followed by a BxBI recombination site, and split rapalog-inducible dimerizable Casp-9. Cells were maintained in D10 (DMEM, 10% fetal bovine serum [FBS], 1% sodium pyruvate, and 1% penicillin/streptomycin). 2 days after transfection, doxycycline (2 µg/ml, Sigma-Aldrich) was added to induce expression of our genes of interest (successful recombination) or the iCasp-9 selection system (no recombination). Successful recombination shifts the iCasp-9 out of frame, thus only cells that have undergone recombination survive, while those that have not will die from iCasp-9-induced apoptosis. 1 day after doxycycline induction, AP1903 (10 nM, MedChemExpress) was added to cause dimerization of Casp9 and selectively kill cells without successful recombination. 1 day after AP1903-Casp9 selection, media was changed back to D10 + doxycycline (2 µg/ml, Sigma-Aldrich) for recovery. 2 days after cells have recovered from all wells, they were mixed and are reseeded to enable normal cell growth. The library cell line therefore represents independent recombination reactions across six wells. Once cells reach confluency, library cells are frozen in 50% FBS and 10% DMSO stocks in aliquots for assays. It was also sequenced to establish a baseline for represented Kir2.1 variants (*Figure 1—figure supplement 2*). Replicate for each assay (surface trafficking and function, see below) represents different aliquots of the same library stock.

## Surface expression cell sorting

Thawed stocks of library cell lines were seeded into a 10 cm dish and media were swapped the following day to D10. Cells were grown and split before confluency to maintain cell health. Media were swapped to D10 + doxycycline (2 μg/ml, Sigma-Aldrich) 2 days prior to the experiment. Cells were detached with 1 ml Accutase (Sigma-Aldrich), spun down and washed three times with FACS buffer (2% FBS, 0.1% $NaN_3$, 1X PBS), incubated for 1 hr rocking at 4°C with a BV421 anti-flag antibody (BD Biosciences), washed twice with FACS buffers, filtered with cell strainer 5 ml tubes (Falcon), covered with aluminum foil, and kept on ice for transfer to the flow cytometry core. Before sorting, 5% of cells were withdrawn for processing and sequencing as a baseline control.

Cells were sorted on a BD FACSAria II P69500132 cell sorter. miRFP670 fluorescence was excited with a 640 nm laser and recorded with a 670/30 nm bandpass filter and 505 nm long-pass filter. BV421 fluorescence was excited using a 405 nm laser. Cells were gated on forward scattering area and side scattering area to find whole cells, forward scattering width, and height to separate single cells, miRFP670 for cells that expressed variants without errors (our library generation results in single base pair deletions that will not have miRFP670 expression because deletions will shift the fluorescent protein out of frame *Coyote-Maestas et al., 2020*), and label for surface-expressed cells. The surface expression label gate boundaries were determined based on unlabeled cells from the same population because controls tend to have nonrepresentative distributions. An example of our gating strategy is depicted in *Figure 1—figure supplement 3*.

Cells were sorted based on surface expression into four populations (miRFP$^{high}$/BV421$^{none}$, miRFP$^{high}$/BV421$^{low}$, miRFP$^{high}$/BV421$^{medium}$, miRFP$^{high}$/BV421$^{high}$). The surface expression experiment was done in duplicate (starting from different aliquots of the library cell line) on separate days for two entirely independent replicates. We collected ~5.1 million cells across both replicates and populations to ensure a greater than 100× coverage for each Kir2.1 variant (*Figure 1—figure supplement 3H*).

## Resting membrane potential cell sorting

Concurrently with preparing samples for surface labeling, another sample of the same cells was prepared for sorting based on RMP. These cells were initially washed with FACS buffer and concentrated in this buffer for cell health prior to sorting. 30 min prior to sorting, cells were resuspended in Tyrode (125 mM NaCl, 2 mM KCl, 3 mM $CaCl_2$, 1 mM $MgCl_2$, 10 mM HEPES, 30 mM glucose, pH 7.3) that contained FLIPR membrane potential dye with a Blue quencher.

Cells were also sorted based on RMP on a BD FACSAria II P69500132 cell sorter. miRFP670 fluorescence was excited with a 640 nm laser and recorded with a 670/30 nm bandpass filter and 505 nm long-pass filter. FLIPR fluorescence was excited using a 488 nm laser and recorded on a 525/50 nm bandpass filter. As before, the same general sorting scheme was used to identify whole single cells based on forward and side scatter and enriched for good quality library members based on miRFP670 fluorescence. An example of our gating strategy is depicted in *Figure 1—figure supplement 4*. Cells were sorted based on RMP into three populations (miRFP$^{high}$/FLIPR$^{Low}$, miRFP$^{high}$/FLIPR$^{medium}$, miRFP$^{high}$/FLIPR$^{high}$). The RMP experiment was done in duplicate (starting from different aliquots of the library cell line) on separate days for two independent replicates. We collected ~3.2 million cells across both replicates and sort populations to ensure a greater than 100× coverage for each Kir2.1 variant (*Figure 1—figure supplement 4H*).

## Sequencing

For both biological replicates, DNA from pre-sort control and sorted cells was extracted with Microprep DNA kits (Zymo Research) and triple-eluted with water. The elute was diluted such that no more than 1.5 μg of DNA was used per PCR reaction and amplified for 20 cycles of PCR using Primestar GXL (Takara Bio), run on a 1% agarose gel, and gel-purified. Primers that bind outside the recombination site ensure leftover plasmid DNA from the original cell line construction step is not amplified. Purified DNA was quantified using Picogreen DNA quantification. Equal amounts (by mass) of each sample were pooled by cell sorting category . Pooled amplicons were prepared for sequencing using the Nextera XT sample preparation workflow and sequenced using Illumina NovaSeq in 2 × 150 bp mode. Source sequencing data is available in the NCBI Sequence Raw Archive (https://www.ncbi.nlm.nih.gov/sra) under accession code PRJNA791691. Read count statistics are listed in *Supplementary file 5*.

## Alignment and calculating variant frequency

Variant frequencies for every single mutation were determined as follows: 150 bp paired-end reads were trimmed to remove standard Illumina adapters (bbmap/resources) using BBDuk with 'mink' at eight bases. Overlapping reads were corrected without merging using BBMerge with the 'ecco' and 'mix' setting. Reads were aligned to the Kir2.1 sequence using BBMap using 'maxindel' at 500 and 'local' alignment. The custom Python script is available at https://github.com/schmidt-lab/KirDMS (*schmidt-lab, 2022*). The resulting SAM file was analyzed for mutation count at each residue position. Since we programmed specific mutations for each codon into the OLS oligo pool that we used for SPINE-mediated variant library synthesis, we could distinguish between expected codons and unexpected codons (predominantly wildtype codons but also sequencing errors). Mutations outside of spine-generated fragments are rare, with 1 bp deletions being the most common (*Coyote-Maestas et al., 2020*). Only programmed (i.e., expected mutations) were counted; the ratio of unexpected/expected mutations for each sample is listed in *Supplementary file 5*. This process of generating a variant list and their counts was repeated for every sample sequenced. For surface fitness assay, this resulted in variant frequencies for 6975–7305 mutations across two replicates of four samples, named from lowest to highest surface expression: 'negative,' 'low,' 'up,' and 'high.' For the function fitness assay, this resulted in variant frequencies for between 7168 and 7320 mutations across two replicates of three samples, named from lowest to highest conductance: 'negative,' 'low,' and 'up.' Coverage of mapped reads is listed in *Figure 2—figure supplement 4* and *Figure 4—figure supplement 3*.

To calculate enrichment of mutants within the surface expression and function assays, we used Enrich2 (*Rubin et al., 2017*) with count data for each mutation as input. Enrich2 calculates fitness scores and their errors based on a fitted weighted-least-squares regression across a series of experiments (e.g., different gating conditions that represent different selection pressures). Each variant's score is defined as the log10-fold transform of slope of the regression line weighted to wildtype frequencies, and the standard deviations within the model account for measurement errors within the sample. This means that positive scores represent enrichment greater than wildtype, while negative scores represent depletion relative to wildtype. Specifically, positive surface expression scores mean higher than wildtype expression, while negative surface scores mean lower than wildtype expression. In function assays, positive scores mean more hyperpolarization relative to wildtype, while negative scores mean less hyperpolarization. Scores were output with a standard error in a .csv file. Positional surface expression and function fitness scores were calculated by taking the mean fitness scores for that experiment across all measured mutations at that position. We were able to collect near-complete datasets for both measured phenotypes. Variant dropout appears to be stochastic; only 20% of variants missing in both surface and function assay were also missing in the stable library cell line baseline (*Figure 1—figure supplement 2*). Some variants were only missing in the baseline library but were detected in phenotyping assay. Taken together, this suggests that dropout occurs at the cell sorting/NGS stages of our workflow. Additional replication, extending sort, and sequencing depth all could help with removing the remaining analysis gaps.

## Inward rectifier phylogenetic alignment and ClinVar mutation assignment

We downloaded all human inward rectifiers (*KCNJ1* [Kir1.1/ROMK1; UniProt P48048], *KCNJ10* [Kir1.2/Kir4.1; UniProt P78508], *KCNJ15* [Kir1.3/Kir4.1; UniProt Q99712], *KCNJ2* [Kir2.1; UniProt P63252], *KCNJ12* [Kir2.2; UniProt Q14500], *KCNJ4* [Kir2.3; UniProt P48050], *KCNJ14* [Kir2.4; UniProt Q9UNX9], *KCNJ18* [Kir2.6; UniProt B7U540], *KCNJ3* [Kir3.1/GIRK1; UniProt P48549], *KCNJ6* [Kir3.2/GIRK2; UniProt P48051], *KCNJ9* [Kir3.3/GIRK3; UniProt Q92806], *KCNJ5* [Kir3.4/GIRK4; UniProt P48544], *KCNJ16* [Kir5.1; UniProt Q9NPI9], *KCNJ8* [Kir6.1; UniProt Q15842], *KCNJ11* [Kir6.2; UniProt Q14654], *KCNJ13* [Kir7.1; UniProt O60928]) and aligned these together using MegaX (*Kumar et al., 2018*). Based on this alignment, we generated a master list of Inward Rectifier numberings to translate residue numbering between Kir homologues. We assigned mutations observed in ClinVar and gnomAD and their pathogenic classification (ClinVar only) to this master alignment (*Supplementary file 1*).

## Acknowledgements

We are grateful for helpful discussions with Anna Gloyn, James Fraser, Gabbriella Estevam, Eric Greene, the DMS crew, and the rest of the Fraser lab. We also want to acknowledge the hard work of the members of the UMN Flow Cytometry Core that enabled us to do these experiments during the COVID-19 pandemic. Rashi Arora especially assisted us in sorting our cells by FACS. We also thank you for taking the time to read our article. This work was supported by the National Institutes of Health (1R01GM136851 to DS) and a University of Minnesota Genome Center Illumina S2 grant. WC-M was supported by a National Science Foundation Graduate Research Fellowship and a Howard Hughes Medical Institute Gilliam Fellowship for Advanced Study.

# Additional information

## Funding

| Funder | Grant reference number | Author |
| --- | --- | --- |
| National Institute of General Medical Sciences | R01GM136851 | Daniel Schmidt |
| Howard Hughes Medical Institute | | Willow Coyote-Maestas |
| Illumina | | Daniel Schmidt |
| National Science Foundation | | Willow Coyote-Maestas |

The funders had no role in study design, data collection and interpretation, or the decision to submit the work for publication.

## Author contributions

Willow Coyote-Maestas, Conceptualization, Data curation, Formal analysis, Investigation, Methodology, Validation, Writing – original draft, Writing – review and editing; David Nedrud, Conceptualization, Data curation, Investigation, Methodology, Software; Yungui He, Investigation; Daniel Schmidt, Conceptualization, Data curation, Formal analysis, Funding acquisition, Investigation, Methodology, Project administration, Software, Supervision, Validation, Visualization, Writing – original draft, Writing – review and editing

## Author ORCIDs

Willow Coyote-Maestas ![ORCID] http://orcid.org/0000-0001-9614-5340
Daniel Schmidt ![ORCID] http://orcid.org/0000-0001-7609-4873

## Decision letter and Author response

Decision letter https://doi.org/10.7554/eLife.76903.sa1
Author response https://doi.org/10.7554/eLife.76903.sa2

# Additional files

## Supplementary files

• Supplementary file 1. Inward rectifier missense variants in ClinVar and gnomAD (as of March 25, 2022). Inward rectifier variants listed in ClinVar and gnomAD along with their expert-reviewed clinical significance (if any). For variants in genes other than *KCNJ2* (i.e., KCNJx), the corresponding Kir2.1 residue, Kir2.1-Flag residue, and amino are listed ('kir21_resno,' 'kir21_FLAG_resno,' 'kir21_resid,' respectively). If the residue in *KCNJx* is equivalent to *KCNJ2*, the 'KCNJx_eq_KCNJ2' column is flagged '1,' and '0' if the residue is different.

• Supplementary file 2. Protein Contact Atlas/residue–residue interactions. Residue interaction as identified in the Protein Contact Atlas (*Kayikci et al., 2018*) for the closed state (PDB 3SPI) and the forced-open state (PDB 6M84) of chicken Kir2.2. Listed for each contact: chains, secondary structure, residue number, amino acid, atom types, chain types, distance (in Angstrom), and contact type.

• Supplementary file 3. Predicted phenotype of KCNJ2 variants of unknown significance. For KCNJ2 variants listed in ClinVar (as of March 25, 2022) with unknown significance or conflicting

interpretation, we predicted mechanism underlying the variants disease phenotype: 'lof_s': loss of surface expression; 'gof_s': gain of surface expression; 'lof_f': loss of function (conduction); 'gof_f': gain of function (conduction).

• Supplementary file 4. Predicted mechanism of pathogenic/conflicting interpretation inward rectifier variants. For all inward rectifier variants listed in ClinVar (as of March 25, 2022) with pathogenic or conflicting interpretation, we predicted mechanism underlying the variants disease phenotype ('predicted_phenotype'): 'lof_s': loss of surface expression; 'gof_s': gain of surface expression; 'lof_f': loss of function (conduction); 'gof_f': gain of function (conduction).

• Supplementary file 5. NextGen Sequencing (NGS) read and alignment statistics. Read statistics for each sequenced variant pool ('sample'). 'Unexpected/expected mutations' is the fraction of mutations that was not encoded in the OLS pool; only expected mutations were used in the analysis.

• Transparent reporting form

• Source data 1. Processed data and scripts to reproduce manuscript figures.

### Data availability

Sequencing data generated in this study have been deposited in the Sequence Raw Archive (https://www.ncbi.nlm.nih.gov/sra) under accession code PRJNA791691. All remaining source data (including processed data and R scripts to reproduce manuscript figures) are included as supplementary information (Source data 1) and are also available at github.com/schmidt-lab/KirDMS (copy archived at swh:1:rev:76cab17b04e2041b3818bcadfc225cd58c9f231f).

The following dataset was generated:

| Author(s) | Year | Dataset title | Dataset URL | Database and Identifier |
|---|---|---|---|---|
| Coyote-Maestas W, Nedrud D, He Y, Schmidt D | 2022 | Multiparametric Deep Mutational Scanning in the Inward Rectifier Kir2.1 | https://www.ncbi.nlm.nih.gov/bioproject/PRJNA791691 | NCBI BioProject, PRJNA791691 |

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
