## [Editor Report]

This manuscript details a new approach to systematically interrogating the relations between the amino acid sequence of ion channels, and their stability, subcellular localization, and function. By applying this approach to an inward-rectifier K^+^-channel, the authors uncover intriguing structural relations between channel regions likely involved in protein stability and those responsible for functional dynamics. Their analysis also offers predictions for the potential pathogenicity of channel variants of unknown significance found in the human population.

---

## [Decision Letter]

**Decision letter after peer review:**

Thank you for submitting your article "Determinants of trafficking, conduction, and disease within a K^+^ channel revealed through multiparametric deep mutational scanning" for consideration by *eLife*. Your article has been reviewed by 3 peer reviewers, one of whom is a member of our Board of Reviewing Editors, and the evaluation has been overseen by Kenton Swartz as the Senior Editor. The following individuals involved in review of your submission have agreed to reveal their identity: Colin G. Nichols (Reviewer #2); Justin Gregory English (Reviewer #3).

The reviewers have discussed their reviews with one another, and the Reviewing Editor has drafted this to help you prepare a revised submission. All three reviewers agree that the manuscript will be of great value to the field, and that the data and analysis presented are of high quality. However, the reviewers identified several points that the authors need to address before publication, which are listed below.

Essential revisions:

1) For Figure S1, it would be informative to also map onto this figure all the currently observed missense mutations in the GnomAD database. Are mutations currently circulating in the population in permissive regions observed later in the study? Are they clustered to particular areas? Looks like majority are in N-term and C-term.

2) For Figure 1C, there does not appear to be a data set sequencing the HEK293T landing pad cell line to demonstrate the extent of library coverage. This is generally important, both for the reader to assess the quality of the initial library as well as to understand the limitations of this method in providing complete coverage maps. I am curious if the cell line coverage landscape would mirror the deficits observed in Supplementary Figure 6 (magneta boxes). While only 7% of the potential data is missing, these gaps appear to be focused in very specific amino acid and receptor domain locations. This data should be made into a heatmap figure (a la figure 2) to demonstrate if the library was uniform or biased toward specific mutational areas. It would be informative to identify how many mutations are under-represented in the library and if this pairs with specific functional deficits. i.e. – are we seeing magenta in Figure 6 because there wasn't sufficient coverage in the initial library to detect them, or are they present, but not measurable?

3) The authors should provide FACS histograms as in Suppl. Figures 2 and 3 for cells expressing only WT channels and for cells without channels – these could be included as an overlay so we can gauge the fluorescence distribution of the negative control cells wherein no channel activity or surface expression is present. It would strengthen the manuscript if other non-functional or surface expression-deficient variants were also included as benchmark, but we leave this up to the authors to decide.

4) For Figure S2 E and F the X-axis is the excitation laser rather than the emission spectra measured. As this signal is referred to as the emission miRPF 67o throughout the paper it may be helpful to change this label to the actual measured unit for the axis, as was done for the Y-axis (BV421). For Figure S3E-F, note in the figure the axis is labeled 640, which is the excitation laser. For consistency it may be best to label this axis as miRFP 670 emission rather than excitation.

5) Figure 2D, Figure 3B, and Figure 5D are too crowded and difficult to decipher. Transparency in some sections or making subunit cartoon representations all the same color could also help. Consider truncating or hiding some elements in the figure to show only regions of interest.

6) For Figure S2 G, these appear to be a sub-sampled statistic of 1% of the total collected population. In the paper (line 634) it was noted that 2.1 million cells were collected, but only 1,000 of each population appear in this statistic table. Can you please also include the final statistics table to evaluate your final sort values. Can you also please note in the legend whether this data is a sub-sample of the published data, a benchmark run, or otherwise. If this gating format is not directly subsampled from the true data collection runs, please re-render these figures using the collected data.

7) Median fitness error to synonymous mutations is reported as "low" for the trafficking defect assay at 7.5%. It is again reported as low for the functional assays at 15.3%. It may be best instead to report the median and deviation of the synonymous and missense datasets and their relative differences at each tail end. Low is a subjective statement here and isn't supported by a positive control measure such as a large N of unmutated channels, run in the same sample, showing similar deviations in these assays as the synonymous mutants. If this data was run, it would be best to include it in the graph for comparison. While not essential, in the future I would encourage the authors to include unmutated controls into their library to assess the impact of synonymous substitutions, and to potentially make single synonymous substitutions at every amino acid position as a reference comparison to the mutants.

8) The authors should include in their discussion some consideration of the "no data" category which generated a consistent number of analysis gaps within certain domains of the channel. How to avoid these in the future? Were these cloning issues? Library construction difficulties with SPINE? Poor coverage of the original cell pool? Could this have been avoided by making multiple cell libraries, sequencing them, and pooling those with compensating coverage?

9) The authors should describe their NGS strategy with much more detail, including more information about the amplicons that they generated for sequencing and how they discriminated between substitutions inside or outside the SPINE-generated variant fragments.

10) The authors should indicate whether the replicates reflect different libraries, different recombined cell libraries, or different stocks of the same library.

11) The "no data" color scheme from Figure 6S should be adopted for the main paper figures. The use of light yellow for no data and white for "0 fold change" makes differentiating these two dataset types difficult visually in these figures. It is important for the reader to be able to easily observe where mutational coverage was lacking in this study.

12) The authors should tone down several observations that are based solely on the DMS screen results. Some of these examples include the increased surface expression of N386E, N386D, T150, Y153, the positive effect in function of N224K and of the residues surrounding the hinge at G176, and importantly, and the predicted phenotypes for the variants of unknown significance. It would further strengthen the manuscript if the authors provided additional experimental validation for a few key examples, either using electrophysiology or the same fluorescence-based assays performed on cell populations expressing single variants, but this is not required.

13) In line 215, the sentence reads: "selectivity filter with glycine significantly increase surface expression." This is difficult to see in Figure 2B, since most mutations in that section of the sequence decrease surface expression. Further, since it is stated to be significant, we believe the authors intended to imply statistical significance? The reader would benefit from a different panel in the figure to highlight the residues within the section of the protein being referenced, as well as information on the statistical test that was run and the p-value.

14) The authors should tone down remarks that involve a distinction between processes that can't be distinguished by their assays, such as co-translational folding and export between different sub-cellular organelles. It needs to be pointed out that the assay is done in an immortalized non-polarized human embryonic kidney cell line. A major caveat to interpretation of trafficking susceptibility is that trafficking constraints may be very different in native cells, and perhaps especially in polarized cells.

15) It would strengthen the manuscript if the authors systematically identified residues engaged in inter vs intra-subunit interactions, and correlated this with the fitness scores for surface expression and function. Otherwise, the observations regarding the role of residues engaging in intra- vs inter-subunit interactions for determining stability and folding or function should be toned down.

16) In line 238, there is a reference to a Fisher's Exact Test, but it's not entirely clear what this test is comparing. We would like to request some clarification of this section, since the current wording suggests that residues involved in gating were analyzed somehow but it's not clear what parameters were used.

17) The authors should systematically compare their per-amino acid variant results with known mutations from the literature, to provide a more critical assessment of their per-variant results. Discrepancies should also be discussed. In Figures 2B and 4C it would be beneficial to place the citation number of the paper corresponding to a previously published defect that matched your observations and border the matching mutation in the heatmap with a unique color identifier so that readers can assess the relative benchmarks and past known mutations and their effects in your assay format more easily. Alternatively, authors could include additional heatmap panels with reduced and simplified content that facilitate comparison between the data in the manuscript and the literature.

18) L189: It is stated that there is agreement between earlier trafficking studies but this statement needs to be validated by a (at least semi-) quantitative comparison, showing a score from the present assay and a score from previous studies, or a Table listing present score and previous results with references, for such residues. This is important given that the authors then talk about establishing a 'global view' of sequence/trafficking relationships. Are there any previous experimental results that do not match? i.e. previous studies showing opposite effects of a given mutation on trafficking? Or on function? While Table S2 compares the present results to clinvar databases, it is unclear whether there are any experimental comparisons.

19) L196: Why is it surprising that pore-lining mutations are tolerant to mutation?

20) L263: What does this sentence mean? The idea that partially functional Kir2.1 will 'eventually' hyperpolarize the cell seems logically wrong. The degree of polarization will depend on the balance of K conductance and other conductances, time dependence is a non sequitur.

21) Paragraph starting l284: This misses any references for the stated mechanistic knowledge. These should be given. The idea that secondary structural elements that couple PIP2 binding to gating can be 'traced' from mutation-sensitive positions is interesting, but far from necessarily correct. This attempt to infer structural connectivity should be made cautiously.

22) Refs 73-75 are 20 years old, and the idea that Kir channels gate by kinking at G176 has been significantly passed over. Gly can be substituted by other residues (DOI: 10.1007/s00249-007-0206-7), and many MD simulations (e.g. DOI: 10.1085/jgp.201912422) show subtle movements along M2 rather than a specific kinking, during gating, as well as highlighting the effects of charge mutations in inducing channel opening. This more nuanced view seems much more consistent with the present data – it is not correct that all mutations reduce function in the present assay. It seems there is improved function for A,T,M,E ,K substitutions.

23) The analysis in Figure 5C conveys no information about the sign and magnitude of the two fitness scores that are being compared; there are multiple possible combinations of surface expression and fitness scores, each implying distinct perturbation mechanisms, that would yield the same calculated difference score. Instead of showing the differences between scores, the authors could show both surface expression and function scores for each position, with a line between the two scores. Different colors could be used for the lines separating the scores, depending on the magnitude of the difference and the sign combination of the two scores. The authors should also discuss if there are positions with a solidly negative surface expression score and a neutral or positive function score – to what extent do these cases raise concern about the accuracy of the function scores? How more reliable are the per-position function scores relative to the per-variant scores, both in regards to what is described above and in general? Granted it is possible that mutations can reduce surface expression without appreciably compromising function, but this should be discussed.

24) Ref 10 is not appropriate for monogenic disease-relevance of Kirs.

25) L47: Neonatal diabetes results from Kir6.2 gain of function. It is confusing to discuss reduced surface expression without noting that this will be a counteracting LOF.

26) On l54 it states that ClinVar lists 163 missense mutations in Kir2.a, but then on l60 it says it reports 144? Something seems wrong.

27) On L66 it is said that there are no large-scale studies of sequence determinants of Kir trafficking and function. The authors should mention Minor et al., (Cell 96:879-91), an early attempt to systematically assess amino acid relevance to structure.

28) L309: What is the important physiological role of GIRK2 here? Presumably you mean Na is an important regulator of GIRK2?

29) In general, it would be best to stick to one nomenclature system, and the system, which names this channel as Kir2.1 (rather than IRK1 or other name) seems best. In this case, GIRK2 should be Kir3.2, ROMK1 should be Kir1.1, etc.

30) Please check the figure numbers and letters that are quoted in the text. We note that lines 279 and 289 reference Figure 4, but it doesn't appear to be referencing the correct panel. For example, line 279, surface and functional data are compared and should reference panels D and E? (Currently references D and F. Similar question on line 289.) Line 631 references the wrong Supplementary Figure. Line 652 also references the wrong Supplementary Figure.

*Reviewer #1 (Recommendations for the authors):*

We would like to recommend specific content changes in order to address the comments raised above. The below suggestions are in the same order to correspond to each point in the public review.

1) The authors should provide FACS histograms as in Suppl. Figures 2 and 3 for cells expressing only WT channels and for cells without channels. Ideally, other non-functional or surface expression-deficient variants should be included as benchmark.

2) The authors should tone down several observations that are based solely on the DMS screen results, and provide additional experimental validation for a few key examples, either using electrophysiology or the same fluorescence-based assays performed on cell populations expressing single variants. Some of these examples include the increased surface expression of N386E, N386D, T150, Y153, the positive effect in function of N224K and of the residues surrounding the hinge at G176, and importantly, at least some of the predicted phenotypes for the variants of unknown significance.

3) The authors should systematically compare their per-amino acid variant results with known mutations from the literature, to provide a more critical assessment of their per-variant results. Discrepancies should also be discussed.

4) It would strengthen the manuscript if the authors systematically identified residues engaged in inter vs intra-subunit interactions, and correlated this with the fitness scores for surface expression and function. Otherwise, the observations regarding the role of residues engaging in intra- vs inter-subunit interactions for determining stability and folding or function should be toned down.

5) The authors should tone down remarks that involve a distinction between processes that can't be distinguished by their assays, such as co-translational folding and export between different sub-cellular organelles.

6) The authors should describe their NGS strategy with detail, including more information about the amplicons that they generated for sequencing and how they discriminated between substitutions inside or outside the SPINE-generated variant fragments. Also, more information about the library should be provided, such as the error rates and error types found after sequencing.

7) The authors should indicate whether the replicates reflect different libraries, different recombined cell libraries, or different stocks of the same library.

8) The analysis in Figure 5C conveys no information about the sign and magnitude of the two fitness scores that are being compared; there are multiple possible combinations of surface expression and fitness scores, each implying distinct perturbation mechanisms, that would yield the same calculated difference score. Instead of showing the differences between scores, the authors could show both surface expression and function scores for each position, with a line between the two scores. Different colors could be used for the lines separating the scores, depending on the magnitude of the difference and the sign combination of the two scores.

9) Related to point (7), the authors should discuss if there are positions with a solidly negative surface expression score and a neutral or positive function score – to what extent do these cases raise concern about the accuracy of the function scores? How more reliable are the per-position function scores relative to the per-variant scores, both in regards to what is described above and in general? Granted it is possible that mutations can reduce surface expression without appreciably compromising function, but this should be discussed.

In paper organization, we would like to note some 'housekeeping' suggestions for the manuscript and figures to increase clarity.

1) In line 215, the sentence reads: "selectivity filter with glycine significantly increase surface expression." This is difficult to see in Figure 2B, since most mutations in that section of the sequence decrease surface expression. Further, since it is stated to be significant, we believe the authors intended to imply statistical significance? The reader would benefit from a different panel in the figure to highlight the residues within the section of the protein being referenced, as well as information on the statistical test that was run and the p-value.

2) In line 238, there is a reference to a Fisher's Exact Test, but it's not entirely clear what this test is comparing. We would like to request some clarification of this section, since the current wording suggests that residues involved in gating were analyzed somehow but it's not clear what parameters were used.

3) Please check the figure numbers and letters that are quoted in the text. We note that lines 279 and 289 reference Figure 4, but it doesn't appear to be referencing the correct panel. For example, line 279, surface and functional data are compared and should reference panels D and E? (Currently references D and F. Similar question on line 289.) Line 631 references the wrong Supplementary Figure. Line 652 also references the wrong Supplementary Figure.

4) Further on figures, could the authors please consider an alternative presentation or more extensive explanation of Figures 6D and E. We respectfully suggest edits are needed to present that point much more forcefully and help the reader understand the significance of the authors' findings. Panels 6 D and E might benefit from more labels on the graph indicating what the dashed lines are. The figure legend and caption don't fully explain the figure, which makes the interesting conclusions very difficult to interpret. Further, it would be helpful if the caption explained briefly what the orange conflicting and green VUS categories are. Are the colors consistent across panels, from A-E? These data are incredibly relevant to the authors' point and different presentation or perhaps a different figure entirely would assist in making this high-impact finding more accessible.

5) Figure 2D, Figure 3B, and Figure 5D are too crowded and difficult to decipher. Transparency in some sections or making subunit cartoon representations all the same color could also help. Consider truncating or hiding some elements in the figure to show only regions of interest.

6) The authors should provide markings (asterisks, arrows) in the heat maps in Figures 2B and 4C, or additional supplementary figures with reduced and simplified content that facilitate discussing observations for specific positions without requiring readers to refer to the heat maps containing data for all variants.

7) In Figure 6C, we suggest a different color for either the variants without predicted mechanism for Kir2.1 (red) and the gain of function (orange), because it is difficult to visually distinguish between them.

*Reviewer #2 (Recommendations for the authors):*

This manuscript reports use of a clever approach to generation of a library of single copy mutants in Kir2.1 and then use of FACS sorting of trafficked and functional mutations to analyze the consequences. The authors are to be congratulated on a data set that is very comprehensive and extremely informative, with important controls in place. The authors have made a strong effort to link their work to previous findings, but the text is written in a discursive style, and the grammatical and typographical errors throughout the manuscript make it difficult to read at times. There are a large number of comments below that should be addressed.

1. L189 It is stated that there is agreement between earlier trafficking studies but this statement needs to be validated by a (at least semi-) quantitative comparison, showing a score from the present assay and a score from previous studies, or a Table listing present score and previous results with references, for such residues. This is important given that the authors then talk about establishing a 'global view' of sequence/trafficking relationships. Are there any previous experimental results that do not match? i.e. previous studies showing opposite effects of a given mutation on trafficking? Or on function? While Table S2 compares the present results to clinvar databases, it is unclear whether there are any experimental comparisons.

2. L196 Why is it surprising that pore-lining mutations are tolerant to mutation?

3. L243-246. This is mere conjectural statement seems unnecessary. Suggest deleting.

4. L263 What does this sentence mean? The idea that partially functional Kir2.1 will 'eventually' hyperpolarize the cell seems logically wrong. The degree of polarization will depend on the balance of K conductance and other conductances, time dependence is a non sequitur.

5. In Figure 5 surface expression was subtracted from function fitness, and while the analysis produces interesting clusters, the real significance is unclear. Rather it seems that the 'functional assessment' is a product of expression and activity of individual channels, and so it seems the analysis should somehow seek to 'divide' the raw functional score by the expression level?

6. Paragraph starting l284. This misses any references for the stated mechanistic knowledge. These should be given. The idea that secondary structural elements that couple PIP2 binding to gating can be 'traced' from mutation-sensitive positions is interesting, but far from necessarily correct. This attempt to infer structural connectivity should be made cautiously.

7. l. 30 paragraph. Refs 73-75 are 20 years old, and the idea that Kir channels gate by kinking at G176 has been significantly passed over. Gly can be substituted by other residues (DOI: 10.1007/s00249-007-0206-7), and many MD simulations (e.g. DOI: 10.1085/jgp.201912422) show subtle movements along M2 rather than a specific kinking, during gating, as well as highlighting the effects of charge mutations in inducing channel opening. This more nuanced view seems much more consistent with the present data – it is not correct that all mutations reduce function in the present assay. It seems there is improved function for A,T,M,E ,K substitutions.

8. It needs to be pointed out that the assay is done in an immortalized non-polarized human embryonic kidney cell line. A major caveat to interpretation of trafficking susceptibility is that trafficking constraints may be very different in native cells, and perhaps especially in polarized cells.

9. There are a many odd typos or grammatical idiosyncracies that need cleaning up (missing articles, oddly inserted 'into', 'to', 'they are', 'and but', 'and' instead of 'the', etc , etc, words throughout the text). The use of speech contractions (e.g. 'We've' for 'We have', l108; 'protein don't' for proteins do not', l458) should all be corrected.

10. l38 Ref 10 is not appropriate for monogenic disease-relevance of Kirs.

11. l47. Neonatal diabetes results from Kir6.2 gain of function. It is confusing to discuss reduced surface expression without noting that this will be a counteracting LOF.

12. On l54 it states that ClinVar lists 163 missense mutations in Kir2.a, but then on l60 it says it reports 144? Something seems wrong.

13. On l66 it is said that there are no large-scale studies of sequence determinants of Kir trafficking and function. The authors should mention Minor et al., (Cell 96:879-91), an early attempt to systematically assess amino acid relevance to structure.

14. l309 What is the important physiological role of GIRK2 here? Presumably you mean Na is an important regulator of GIRK2?

15. In general, it would be best to stick to one nomenclature system, and the system, which names this channel as Kir2.1 (rather than IRK1 or other name) seems best. In this case, GIRK2 should be Kir3.2, ROMK1 should be Kir1.1, etc.

16. Paragraph beginning l2438 belongs in discussion.

17. There are weirdly incomplete references e.g, refs 97-99.

*Reviewer #3 (Recommendations for the authors):*

1. For Figure S1, it would be informative to also map onto this figure all the currently observed missense mutations in the GnomAD database. Are mutations currently circulating in the population in permissive regions observed later in the study? Are they clustered to particular areas? Looks like majority are in N-term and C-term.

2. For Figure 1C, there does not appear to be a data set sequencing the HEK293T landing pad cell line to demonstrate the extent of library coverage. This is generally important, both for the reader to assess the quality of the initial library as well as to understand the limitations of this method in providing complete coverage maps. I am curious if the cell line coverage landscape would mirror the deficits observed in Supplementary Figure 6 (magneta boxes). While only 7% of the potential data is missing, these gaps appear to be focused in very specific amino acid and receptor domain locations. This data should be made into a heatmap figure (a la figure 2) to demonstrate if the library was uniform or biased toward specific mutational areas. It would be informative to identify how many mutations are under-represented in the library and if this pairs with specific functional deficits. i.e. – are we seeing magenta in Figure 6 because there wasn't sufficient coverage in the initial library to detect them, or are they present, but not measurable?

3. In Figure 2b it would be beneficial to place the citation number of the paper corresponding to a previously published defect that matched your observations and border the matching mutation in the heatmap with a unique color identifier so that readers can assess the relative benchmarks and past known mutations and their effects in your assay format more easily.

4. For Figure S2 E and F the X-axis is the excitation laser rather than the emission spectra measured. As this signal is referred to as the emission miRPF 67o throughout the paper it may be helpful to change this label to the actual measured unit for the axis, as was done for the Y-axis (BV421).

5. For Figure S2 G, these appear to be a sub-sampled statistic of 1% of the total collected population. In the paper (line 634) it was noted that 2.1 million cells were collected, but only 1,000 of each population appear in this statistic table. Can you please also include the final statistics table to evaluate your final sort values. Can you also please note in the legend whether this data is a sub-sample of the published data, a benchmark run, or otherwise. If this gating format is not directly subsampled from the true data collection runs, please re-render these figures using the collected data.

6. Median fitness error to synonymous mutations is reported as "low" for the trafficking defect assay at 7.5%. It is again reported as low for the functional assays at 15.3%. It may be best instead to report the median and deviation of the synonymous and missense datasets and their relative differences at each tail end. Low is a subjective statement here and isn't supported by a positive control measure such as a large N of unmutated channels, run in the same sample, showing similar deviations in these assays as the synonymous mutants. If this data was run, it would be best to include it in the graph for comparison. While not essential, in the future I would encourage the authors to include unmutated controls into their library to assess the impact of synonymous substitutions, and to potentially make single synonymous substitutions at every amino acid position as a reference comparison to the mutants.

7. For Figure S3E-F, please show the FLIPR fluorescence of the miRFP 670 negative population as an overlay in F (dotted line of peaks over the red of P4) so we can gauge the fluorescence distribution of the negative control cells wherein no channel activity is present. Is this how "FLIPR-" was determined? Also, note in the figure the axis is labeled 640, which is the excitation laser. For consistency it may be best to label this axis as miRFP 670 emission rather than excitation.

8. The authors should include in their discussion some consideration of the "no data" category which generated a consistent number of analysis gaps within certain domains of the channel. How to avoid these in the future? Were these cloning issues? Library construction difficulties with SPINE? Poor coverage of the original cell pool? Could this have been avoided by making multiple cell libraries, sequencing them, and pooling those with compensating coverage?

9. The "no data" color scheme from Figure 6S should be adopted for the main paper figures. The use of light yellow for no data and white for "0 fold change" makes differentiating these two dataset types difficult visually in these figures. It is important for the reader to be able to easily observe where mutational coverage was lacking in this study.

---

## [Author Response]

Essential revisions:1) For Figure S1, it would be informative to also map onto this figure all the currently observed missense mutations in the GnomAD database. Are mutations currently circulating in the population in permissive regions observed later in the study? Are they clustered to particular areas? Looks like majority are in N-term and C-term.

We have updated reported missense mutations (now Figure 1—figure supplement 1) to include a more recent release of ClinVar (as of 3/25/2022) and added the gnomAD data (v.2.1.1). gnomADonly variants do not appear to cluster in a specific area of Kir2.1.

2) For Figure 1C, there does not appear to be a data set sequencing the HEK293T landing pad cell line to demonstrate the extent of library coverage. This is generally important, both for the reader to assess the quality of the initial library as well as to understand the limitations of this method in providing complete coverage maps. I am curious if the cell line coverage landscape would mirror the deficits observed in Supplementary Figure 6 (magneta boxes). While only 7% of the potential data is missing, these gaps appear to be focused in very specific amino acid and receptor domain locations. This data should be made into a heatmap figure (a la figure 2) to demonstrate if the library was uniform or biased toward specific mutational areas. It would be informative to identify how many mutations are under-represented in the library and if this pairs with specific functional deficits. i.e. – are we seeing magenta in Figure 6 because there wasn't sufficient coverage in the initial library to detect them, or are they present, but not measurable?

We have added these data as new supplementary figures (Figure 1—figure supplement 2 and Figure 2-figure supplement 1) and analysis discussed in the main results section. Some missing data localizes to the “seams” of SPINE-generated fragments from which the Kir2.1 ORF is assembled. This is a known issue with SPINE and may be addressed by using overlapping fragment assembly. Nevertheless, our data comparing missing data in the stable library cell line, surface expression assay, and function assay suggests that variant dropout is stochastic occurring at the cell sorting / NGS stages of our workflow. Additional replication, extending sort and sequencing depth all could help with removing the remaining analysis gaps. We have added this data and discussion.

3) The authors should provide FACS histograms as in Suppl. Figures 2 and 3 for cells expressing only WT channels and for cells without channels – these could be included as an overlay so we can gauge the fluorescence distribution of the negative control cells wherein no channel activity or surface expression is present. It would strengthen the manuscript if other non-functional or surface expression-deficient variants were also included as benchmark, but we leave this up to the authors to decide.

We have added these histograms; both include WT Kir2.1 channels (Figure 1—figure supplement 3 and Figure 1—figure supplement 4). For surface expression assays, we have included a negative control HEK cell line expressing Kv1.3, which lacks the extracellular epitope and therefore will not be labelled by the FLAG antibody. For the function assays, we have included V302M as a negative control, which abolishes gating but does not affect surface expression (Ma […] Welling 2007).

4) For Figure S2 E and F the X-axis is the excitation laser rather than the emission spectra measured. As this signal is referred to as the emission miRPF 67o throughout the paper it may be helpful to change this label to the actual measured unit for the axis, as was done for the Y-axis (BV421). For Figure S3E-F, note in the figure the axis is labeled 640, which is the excitation laser. For consistency it may be best to label this axis as miRFP 670 emission rather than excitation.

We have updated all figures to say that fluorescence intensity at the respective emission wavelength of the fluorophore is shown.

5) Figure 2D, Figure 3B, and Figure 5D are too crowded and difficult to decipher. Transparency in some sections or making subunit cartoon representations all the same color could also help. Consider truncating or hiding some elements in the figure to show only regions of interest.

We have simplified these figures, and several others, to make them easier to understand.

6) For Figure S2 G, these appear to be a sub-sampled statistic of 1% of the total collected population. In the paper (line 634) it was noted that 2.1 million cells were collected, but only 1,000 of each population appear in this statistic table. Can you please also include the final statistics table to evaluate your final sort values. Can you also please note in the legend whether this data is a sub-sample of the published data, a benchmark run, or otherwise. If this gating format is not directly subsampled from the true data collection runs, please re-render these figures using the collected data.

We have included count statistics for all collected cell population in these Supplementary Figures (Figure 1—figure supplement 3 and Figure 1—figure supplement 4). We have also updated the figure to reflect that gating examples are based on a sub-sample of the collected cell population on which this study is based.

7) Median fitness error to synonymous mutations is reported as "low" for the trafficking defect assay at 7.5%. It is again reported as low for the functional assays at 15.3%. It may be best instead to report the median and deviation of the synonymous and missense datasets and their relative differences at each tail end. Low is a subjective statement here and isn't supported by a positive control measure such as a large N of unmutated channels, run in the same sample, showing similar deviations in these assays as the synonymous mutants. If this data was run, it would be best to include it in the graph for comparison. While not essential, in the future I would encourage the authors to include unmutated controls into their library to assess the impact of synonymous substitutions, and to potentially make single synonymous substitutions at every amino acid position as a reference comparison to the mutants.

We have added these metrics and removed the less quantitative statements. Due the specific implementation of our genotype/phenotype linkage workflow (fragmentation of amplicons and short-read NextGen Sequencing), it is not feasible to include unmutated controls in our libraries. For this reason, we used synonymous mutations to assess wildtype fitness (a common approach in the field). We will consider switching to a combination of long-read sequencing (Sequel) and unique molecular identifiers for future studies.

8) The authors should include in their discussion some consideration of the "no data" category which generated a consistent number of analysis gaps within certain domains of the channel. How to avoid these in the future? Were these cloning issues? Library construction difficulties with SPINE? Poor coverage of the original cell pool? Could this have been avoided by making multiple cell libraries, sequencing them, and pooling those with compensating coverage?

Our data comparing missing data in the stable library cell line, surface expression assay, and function assay suggests that variant dropout is stochastic occurring at the cell sorting / NGS stages of our workflow (new Figure 2—figure supplement 1). Additional replication, extending sort and sequencing depth all could help with removing the remaining analysis gaps. We have added this data and discussion.

9) The authors should describe their NGS strategy with much more detail, including more information about the amplicons that they generated for sequencing and how they discriminated between substitutions inside or outside the SPINE-generated variant fragments.

We have edited this section in the Methods description:

– We describe how amplicon libraries are prepared (limited cycle PCR of purified genomic DNA) followed by NexteraXT tagmentation for Illumina Sequencing.

– We describe how reads are analyzed to generate a list of variants and their counts. We are making the analysis script (written in Python) available at https://github.com/schmidtlab/KirDMS. Because we use OLS oligo pools with specific programmed mutations, we can distinguish between expected mutations at each codon and unexpected mutations (wildtype codon, sequencing errors etc.). Our analysis is based on expected mutation counts only.

10) The authors should indicate whether the replicates reflect different libraries, different recombined cell libraries, or different stocks of the same library.

We have added this information. Replicates reflect different aliquots of the same recombined libraries that were frozen down, then thawed, and expanded less than one week prior to experiments. Each biology replicate was done on a different day.

11) The "no data" color scheme from Figure 6S should be adopted for the main paper figures. The use of light yellow for no data and white for "0 fold change" makes differentiating these two dataset types difficult visually in these figures. It is important for the reader to be able to easily observe where mutational coverage was lacking in this study.

We have adapted the magenta = “no data” scheme for figures that illustrate fitness errors and dark grey (grey60) in Figures 2 and 4. Magenta would not work for the latter two figures; other primary colors are too distracting. In our view, a dark gray tone makes it easy to see where data is missing, and this is a standard in the field (e.g. https://elifesciences.org/articles/56707 and https://elifesciences.org/articles/15802).

12) The authors should tone down several observations that are based solely on the DMS screen results. Some of these examples include the increased surface expression of N386E, N386D, T150, Y153, the positive effect in function of N224K and of the residues surrounding the hinge at G176, and importantly, and the predicted phenotypes for the variants of unknown significance. It would further strengthen the manuscript if the authors provided additional experimental validation for a few key examples, either using electrophysiology or the same fluorescence-based assays performed on cell populations expressing single variants, but this is not required.

We are confident in the premise that our assays for surface expression and function provides sufficient sensitivity, dynamic range, and fidelity. This is borne out of prior work (e.g., CoyoteMaestas Nat Comm 2019) and controls included in this study (e.g., Kv1.3 control for surface expression; V302M as a negative control for function).

That said, we agree that additional experimental for single variants could be useful to further bolster this premise. However, finding the right balance between how many variants to test and which metrics to use for comparison (for function – open probability, selectivity, etc.) is not immediately obvious.

Furthermore, since we are using single copy stable cell lines (each cell expresses only one variant) there is not conceptual difference between performing the same fluorescence assay on cell populations expressing a single variant vs. multiple variants. The former is simply a subsample of the latter; we would not expect the distribution of fitness scores to change, only the measurement error would improve.

We have therefore addressed this request in the following ways:

– We have edited the text to make it clear that we are only reporting fitness scores measured by our assays.

– Considering noisy data (in particular for function), we refrain drawing conclusion from single variant but instead focus on themes emerging when taking several variant data points in aggregate.

– We have included additional comparison to prior studies that reported variant effect on surface expression and/or function. We find that there is very strong correlation between our results and prior data, which adds confidence to our result and interpretations.

– In several instance (e.g., surface expression of T150 and Y153 variants, we have added additional analysis and illustration).

– We have removed our speculative statement about N224K.

– We edited the text to emphasize that our interpretation of lower glycine hinge mutations is speculative at this point and requires more validation.

– With respect to predicted phenotypes for variants of unknown significance, we respectfully point out that “predicted” implies uncertainty in the phenotype we are assigning. Even if we were to make individual variants, test them by electrophysiology, and find the same phenotype that our high-throughput assay reported, we would not necessarily have more confidence in assigning clinical significance. That is because of the heavy focus medical genetics and genomics places on case evidence (i.e., finding these variants in patient with disease).

13) In line 215, the sentence reads: "selectivity filter with glycine significantly increase surface expression." This is difficult to see in Figure 2B, since most mutations in that section of the sequence decrease surface expression. Further, since it is stated to be significant, we believe the authors intended to imply statistical significance? The reader would benefit from a different panel in the figure to highlight the residues within the section of the protein being referenced, as well as information on the statistical test that was run and the p-value.

We have added additional analysis and illustration (Figure 3B-D). We find that selectivity filter residues are more tolerant to mutations (when measuring surface expression) than the surrounding structural elements (TM1 and Pore Helix).

14) The authors should tone down remarks that involve a distinction between processes that can't be distinguished by their assays, such as co-translational folding and export between different sub-cellular organelles. It needs to be pointed out that the assay is done in an immortalized non-polarized human embryonic kidney cell line. A major caveat to interpretation of trafficking susceptibility is that trafficking constraints may be very different in native cells, and perhaps especially in polarized cells.

We have rephrased these sections and softened these remarks. It is correct that we currently cannot distinguish between effects on early folding events and later trafficking. However, by using our experimentally measured surface fitness score in a region that is equivalent to a “cotranslationally folded biogenic unit” in other K^+^ channels, we use inductive reasoning to propose that a similar cotranslationally-folded biogenic unit exists in Kir2.1. We now go on to say that further assay development to separate variant effect on folding vs. export and trafficking between different cell organelles would be very useful.

We thank the reviewer reminding us of context dependence when study ion channel trafficking and biophysics (endogenous vs. overexpression, primary culture, cell lines, oocytes, etc.). We have edited the discussion to point out the caveats that come with performing these assays in HEK293.

15) It would strengthen the manuscript if the authors systematically identified residues engaged in inter vs intra-subunit interactions, and correlated this with the fitness scores for surface expression and function. Otherwise, the observations regarding the role of residues engaging in intra- vs inter-subunit interactions for determining stability and folding or function should be toned down.

We have added this analysis and new figure panels (Figure 5—figure supplement 1). For CTD residues, we can show that intra-subunit contacts are the driver of surface-expression, while intersubunit contact drive function fitness.

16) In line 238, there is a reference to a Fisher's Exact Test, but it's not entirely clear what this test is comparing. We would like to request some clarification of this section, since the current wording suggests that residues involved in gating were analyzed somehow but it's not clear what parameters were used.

We have clarified what this test is comparing: the odds that a loss of surface expression (LoS) vs. a loss of function (LoF) is predicted for experimentally measured variants vs. variants in ClinVar (see Figure 7—figure supplement 1). The odds are significantly higher (odds ratio LoS/LoF 4.6, p-value 0.0001145), meaning that loss of surface expression variants are underrepresented in ClinVar.

17) The authors should systematically compare their per-amino acid variant results with known mutations from the literature, to provide a more critical assessment of their per-variant results. Discrepancies should also be discussed. In Figures 2B and 4C it would be beneficial to place the citation number of the paper corresponding to a previously published defect that matched your observations and border the matching mutation in the heatmap with a unique color identifier so that readers can assess the relative benchmarks and past known mutations and their effects in your assay format more easily. Alternatively, authors could include additional heatmap panels with reduced and simplified content that facilitate comparison between the data in the manuscript and the literature.

We have chosen three comparison studies that allow us to make quantitative comparisons:

Ma et al., 2011, *Cell* Comparing variant effect on Golgi export and surface expression

Dart et al., 1998, *JGP* Accessibility of selectivity filter-lining residues to Ag^+^

Minor et al., 1999, *Cell* Allowed substitutions in functional Kir2.1 channels

These comparisons are conveyed in a new Figure 6 and under the result subheading “Surface and Function DMS results agree with prior studies”. Our data are in strong agreement with prior these prior studies.

18) L189: It is stated that there is agreement between earlier trafficking studies but this statement needs to be validated by a (at least semi-) quantitative comparison, showing a score from the present assay and a score from previous studies, or a Table listing present score and previous results with references, for such residues. This is important given that the authors then talk about establishing a 'global view' of sequence/trafficking relationships. Are there any previous experimental results that do not match? i.e. previous studies showing opposite effects of a given mutation on trafficking? Or on function? While Table S2 compares the present results to clinvar databases, it is unclear whether there are any experimental comparisons.

See our response to the comment above.

19) L196: Why is it surprising that pore-lining mutations are tolerant to mutation?

We have removed the word “surprisingly”.

20) L263: What does this sentence mean? The idea that partially functional Kir2.1 will 'eventually' hyperpolarize the cell seems logically wrong. The degree of polarization will depend on the balance of K conductance and other conductances, time dependence is a non sequitur.

We have corrected this sentence removing the impression that we are talking about a nonequilibrium process.

21) Paragraph starting l284: This misses any references for the stated mechanistic knowledge. These should be given. The idea that secondary structural elements that couple PIP2 binding to gating can be 'traced' from mutation-sensitive positions is interesting, but far from necessarily correct. This attempt to infer structural connectivity should be made cautiously.

We have edited this section to highlight from which prior studies our statements of mechanistic knowledge are drawn. We use more cautious phrasing about potential coupling between subunit interfaces.

22) Refs 73-75 are 20 years old, and the idea that Kir channels gate by kinking at G176 has been significantly passed over. Gly can be substituted by other residues (DOI: 10.1007/s00249-007-0206-7), and many MD simulations (e.g. DOI: 10.1085/jgp.201912422) show subtle movements along M2 rather than a specific kinking, during gating, as well as highlighting the effects of charge mutations in inducing channel opening. This more nuanced view seems much more consistent with the present data – it is not correct that all mutations reduce function in the present assay. It seems there is improved function for A,T,M,E ,K substitutions.

The thank the reviewer for bringing this to our attention; we have edited this section accordingly.

23) The analysis in Figure 5C conveys no information about the sign and magnitude of the two fitness scores that are being compared; there are multiple possible combinations of surface expression and fitness scores, each implying distinct perturbation mechanisms, that would yield the same calculated difference score. Instead of showing the differences between scores, the authors could show both surface expression and function scores for each position, with a line between the two scores. Different colors could be used for the lines separating the scores, depending on the magnitude of the difference and the sign combination of the two scores. The authors should also discuss if there are positions with a solidly negative surface expression score and a neutral or positive function score – to what extent do these cases raise concern about the accuracy of the function scores? How more reliable are the per-position function scores relative to the per-variant scores, both in regards to what is described above and in general? Granted it is possible that mutations can reduce surface expression without appreciably compromising function, but this should be discussed.

We have created a new version of this figure along the lines of what the reviewer is suggesting. We have added a clarification that explain the apparent discordance of low surface fitness and neutral function fitness: it is due to the difference in measurement dynamic range between surface expression and function assays. The function assay is measuring steady-state RMP and fully functional Kir2.1, even at lower surface expression level may hyperpolarize the cell, giving the appearance of a neutral phenotype.

24) Ref 10 is not appropriate for monogenic disease-relevance of Kirs.

We have updated these references.

25) L47: Neonatal diabetes results from Kir6.2 gain of function. It is confusing to discuss reduced surface expression without noting that this will be a counteracting LOF.

We have edited this sentence to clarify that neonatal diabetes is caused by GOF mutations in Kir6.2.

26) On l54 it states that ClinVar lists 163 missense mutations in Kir2.a, but then on l60 it says it reports 144? Something seems wrong.

ClinVar continuously is updated with new information; we had overlooked one instance referring to an older version. In this revised version, we are using ClinVar data pull on March 25, 2022 and also gnomAD data pulled on the same day.

27) On L66 it is said that there are no large-scale studies of sequence determinants of Kir trafficking and function. The authors should mention Minor et al., (Cell 96:879-91), an early attempt to systematically assess amino acid relevance to structure.

We are now mentioning this early study. We are also using it as one of the validation datasets for a quantitative comparison to earlier literature datasets.

28) L309: What is the important physiological role of GIRK2 here? Presumably you mean Na is an important regulator of GIRK2?

We have removed this paragraph. It seemed superfluous to the point we are trying to make.

29) In general, it would be best to stick to one nomenclature system, and the system, which names this channel as Kir2.1 (rather than IRK1 or other name) seems best. In this case, GIRK2 should be Kir3.2, ROMK1 should be Kir1.1, etc.

We have changed everything to the KirX.Y nomenclature.

30) Please check the figure numbers and letters that are quoted in the text. We note that lines 279 and 289 reference Figure 4, but it doesn't appear to be referencing the correct panel. For example, line 279, surface and functional data are compared and should reference panels D and E? (Currently references D and F. Similar question on line 289.) Line 631 references the wrong Supplementary Figure. Line 652 also references the wrong Supplementary Figure.

We have fixed these errors.

Reviewer #1 (Recommendations for the authors):We would like to recommend specific content changes in order to address the comments raised above. The below suggestions are in the same order to correspond to each point in the public review.1) The authors should provide FACS histograms as in Suppl. Figures 2 and 3 for cells expressing only WT channels and for cells without channels. Ideally, other non-functional or surface expression-deficient variants should be included as benchmark.

This is addressed by Essential Revision #3.

2) The authors should tone down several observations that are based solely on the DMS screen results, and provide additional experimental validation for a few key examples, either using electrophysiology or the same fluorescence-based assays performed on cell populations expressing single variants. Some of these examples include the increased surface expression of N386E, N386D, T150, Y153, the positive effect in function of N224K and of the residues surrounding the hinge at G176, and importantly, at least some of the predicted phenotypes for the variants of unknown significance.

This is addressed by Essential Revision #12.

3) The authors should systematically compare their per-amino acid variant results with known mutations from the literature, to provide a more critical assessment of their per-variant results. Discrepancies should also be discussed.

This is addressed by Essential Revision #17.

4) It would strengthen the manuscript if the authors systematically identified residues engaged in inter vs intra-subunit interactions, and correlated this with the fitness scores for surface expression and function. Otherwise, the observations regarding the role of residues engaging in intra- vs inter-subunit interactions for determining stability and folding or function should be toned down.

This is addressed by Essential Revision #15.

5) The authors should tone down remarks that involve a distinction between processes that can't be distinguished by their assays, such as co-translational folding and export between different sub-cellular organelles.

This is addressed by Essential Revision #14.

6) The authors should describe their NGS strategy with detail, including more information about the amplicons that they generated for sequencing and how they discriminated between substitutions inside or outside the SPINE-generated variant fragments. Also, more information about the library should be provided, such as the error rates and error types found after sequencing.

This is addressed by Essential Revision #9.

7) The authors should indicate whether the replicates reflect different libraries, different recombined cell libraries, or different stocks of the same library.

This is addressed by Essential Revision #10.

8) The analysis in Figure 5C conveys no information about the sign and magnitude of the two fitness scores that are being compared; there are multiple possible combinations of surface expression and fitness scores, each implying distinct perturbation mechanisms, that would yield the same calculated difference score. Instead of showing the differences between scores, the authors could show both surface expression and function scores for each position, with a line between the two scores. Different colors could be used for the lines separating the scores, depending on the magnitude of the difference and the sign combination of the two scores.

This is addressed by Essential Revision #23.

9) Related to point (7), the authors should discuss if there are positions with a solidly negative surface expression score and a neutral or positive function score – to what extent do these cases raise concern about the accuracy of the function scores? How more reliable are the per-position function scores relative to the per-variant scores, both in regards to what is described above and in general? Granted it is possible that mutations can reduce surface expression without appreciably compromising function, but this should be discussed.

This is addressed by Essential Revision #23.

In paper organization, we would like to note some 'housekeeping' suggestions for the manuscript and figures to increase clarity.1) In line 215, the sentence reads: "selectivity filter with glycine significantly increase surface expression." This is difficult to see in Figure 2B, since most mutations in that section of the sequence decrease surface expression. Further, since it is stated to be significant, we believe the authors intended to imply statistical significance? The reader would benefit from a different panel in the figure to highlight the residues within the section of the protein being referenced, as well as information on the statistical test that was run and the p-value.

This is addressed by Essential Revision #13.

2) In line 238, there is a reference to a Fisher's Exact Test, but it's not entirely clear what this test is comparing. We would like to request some clarification of this section, since the current wording suggests that residues involved in gating were analyzed somehow but it's not clear what parameters were used.

This is addressed by Essential Revision #16.

3) Please check the figure numbers and letters that are quoted in the text. We note that lines 279 and 289 reference Figure 4, but it doesn't appear to be referencing the correct panel. For example, line 279, surface and functional data are compared and should reference panels D and E? (Currently references D and F. Similar question on line 289.) Line 631 references the wrong Supplementary Figure. Line 652 also references the wrong Supplementary Figure.

This is addressed by Essential Revision #30.

4) Further on figures, could the authors please consider an alternative presentation or more extensive explanation of Figures 6D and E. We respectfully suggest edits are needed to present that point much more forcefully and help the reader understand the significance of the authors' findings. Panels 6 D and E might benefit from more labels on the graph indicating what the dashed lines are. The figure legend and caption don't fully explain the figure, which makes the interesting conclusions very difficult to interpret. Further, it would be helpful if the caption explained briefly what the orange conflicting and green VUS categories are. Are the colors consistent across panels, from A-E? These data are incredibly relevant to the authors' point and different presentation or perhaps a different figure entirely would assist in making this high-impact finding more accessible.

We have updated this Figure (now: Figure 7). We simplified panels D and E. We have added an explanation for 'conflicting' vs. 'VUS' – essentially, conflicting interpretation vs. unknown significance. Colors are consistent within panel A, within panel C, and within panels D & E, but not across panels. We have chose visually distinct color schemes to avoid confusion.

5) Figure 2D, Figure 3B, and Figure 5D are too crowded and difficult to decipher. Transparency in some sections or making subunit cartoon representations all the same color could also help. Consider truncating or hiding some elements in the figure to show only regions of interest.

This is addressed by Essential Revision #5.

6) The authors should provide markings (asterisks, arrows) in the heat maps in Figures 2B and 4C, or additional supplementary figures with reduced and simplified content that facilitate discussing observations for specific positions without requiring readers to refer to the heat maps containing data for all variants.

We have attempted adding asterisk and other visual aids to highlight specific position mentioned in the text, but this added clutter to an already complicated heatmap. Considering noisy data (in particular for function), we found that drawing from conclusion from several data points in aggregate is better than focusing on single variants. Where possible, we have added dedicated panels with fitness score mapped onto the Kir2.2 structure to provide visual context.

7) In Figure 6C, we suggest a different color for either the variants without predicted mechanism for Kir2.1 (red) and the gain of function (orange), because it is difficult to visually distinguish between them.

We updated this figure (now: Figure 7C) and chose visually distinct color schemes to avoid confusion.

Reviewer #2 (Recommendations for the authors):This manuscript reports use of a clever approach to generation of a library of single copy mutants in Kir2.1 and then use of FACS sorting of trafficked and functional mutations to analyze the consequences. The authors are to be congratulated on a data set that is very comprehensive and extremely informative, with important controls in place. The authors have made a strong effort to link their work to previous findings, but the text is written in a discursive style, and the grammatical and typographical errors throughout the manuscript make it difficult to read at times. There are a large number of comments below that should be addressed.1. L189 It is stated that there is agreement between earlier trafficking studies but this statement needs to be validated by a (at least semi-) quantitative comparison, showing a score from the present assay and a score from previous studies, or a Table listing present score and previous results with references, for such residues. This is important given that the authors then talk about establishing a 'global view' of sequence/trafficking relationships. Are there any previous experimental results that do not match? i.e. previous studies showing opposite effects of a given mutation on trafficking? Or on function? While Table S2 compares the present results to clinvar databases, it is unclear whether there are any experimental comparisons.

This is addressed by Essential Revision #18.

2. L196 Why is it surprising that pore-lining mutations are tolerant to mutation?

This is addressed by Essential Revision #19.

3. L243-246. This is mere conjectural statement seems unnecessary. Suggest deleting.

This statement relates to the major motivation for using perturbation as a coarse grain high-throughput approach to annotate protein structure – a major focus of our ongoing research. We respectfully request to leave this statement as is, even if conjecture at this point.

4. L263 What does this sentence mean? The idea that partially functional Kir2.1 will 'eventually' hyperpolarize the cell seems logically wrong. The degree of polarization will depend on the balance of K conductance and other conductances, time dependence is a non sequitur.

This is addressed by Essential Revision #20.

5. In Figure 5 surface expression was subtracted from function fitness, and while the analysis produces interesting clusters, the real significance is unclear. Rather it seems that the 'functional assessment' is a product of expression and activity of individual channels, and so it seems the analysis should somehow seek to 'divide' the raw functional score by the expression level?

This is addressed by Essential Revision #23.

6. Paragraph starting l284. This misses any references for the stated mechanistic knowledge. These should be given. The idea that secondary structural elements that couple PIP2 binding to gating can be 'traced' from mutation-sensitive positions is interesting, but far from necessarily correct. This attempt to infer structural connectivity should be made cautiously.

This is addressed by Essential Revision #21.

7. l. 30 paragraph. Refs 73-75 are 20 years old, and the idea that Kir channels gate by kinking at G176 has been significantly passed over. Gly can be substituted by other residues (DOI: 10.1007/s00249-007-0206-7), and many MD simulations (e.g. DOI: 10.1085/jgp.201912422) show subtle movements along M2 rather than a specific kinking, during gating, as well as highlighting the effects of charge mutations in inducing channel opening. This more nuanced view seems much more consistent with the present data – it is not correct that all mutations reduce function in the present assay. It seems there is improved function for A,T,M,E ,K substitutions.

This is addressed by Essential Revision #22.

8. It needs to be pointed out that the assay is done in an immortalized non-polarized human embryonic kidney cell line. A major caveat to interpretation of trafficking susceptibility is that trafficking constraints may be very different in native cells, and perhaps especially in polarized cells.

This is addressed by Essential Revision #14.

9. There are a many odd typos or grammatical idiosyncracies that need cleaning up (missing articles, oddly inserted 'into', 'to', 'they are', 'and but', 'and' instead of 'the', etc , etc, words throughout the text). The use of speech contractions (e.g. 'We've' for 'We have', l108; 'protein don't' for proteins do not', l458) should all be corrected.

We have fixed these.

10. l38 Ref 10 is not appropriate for monogenic disease-relevance of Kirs.

This is addressed by Essential Revision #24.

11. l47. Neonatal diabetes results from Kir6.2 gain of function. It is confusing to discuss reduced surface expression without noting that this will be a counteracting LOF.

This is addressed by Essential Revision #25.

12. On l54 it states that ClinVar lists 163 missense mutations in Kir2.a, but then on l60 it says it reports 144? Something seems wrong.

This is addressed by Essential Revision #26.

13. On l66 it is said that there are no large-scale studies of sequence determinants of Kir trafficking and function. The authors should mention Minor et al., (Cell 96:879-91), an early attempt to systematically assess amino acid relevance to structure.

This is addressed by Essential Revision #27.

14. l309 What is the important physiological role of GIRK2 here? Presumably you mean Na is an important regulator of GIRK2?

This is addressed by Essential Revision #28.

15. In general, it would be best to stick to one nomenclature system, and the system, which names this channel as Kir2.1 (rather than IRK1 or other name) seems best. In this case, GIRK2 should be Kir3.2, ROMK1 should be Kir1.1, etc.

This is addressed by Essential Revision #29.

16. Paragraph beginning l2438 belongs in discussion.

This is addressed in Essential Revision. This is addressed by our response to reviewer comment 3.

17. There are weirdly incomplete references e.g, refs 97-99.

Fixed.

Reviewer #3 (Recommendations for the authors):1. For Figure S1, it would be informative to also map onto this figure all the currently observed missense mutations in the GnomAD database. Are mutations currently circulating in the population in permissive regions observed later in the study? Are they clustered to particular areas? Looks like majority are in N-term and C-term.

This is addressed by Essential Revision #1.

2. For Figure 1C, there does not appear to be a data set sequencing the HEK293T landing pad cell line to demonstrate the extent of library coverage. This is generally important, both for the reader to assess the quality of the initial library as well as to understand the limitations of this method in providing complete coverage maps. I am curious if the cell line coverage landscape would mirror the deficits observed in Supplementary Figure 6 (magneta boxes). While only 7% of the potential data is missing, these gaps appear to be focused in very specific amino acid and receptor domain locations. This data should be made into a heatmap figure (a la figure 2) to demonstrate if the library was uniform or biased toward specific mutational areas. It would be informative to identify how many mutations are under-represented in the library and if this pairs with specific functional deficits. i.e. – are we seeing magenta in Figure 6 because there wasn't sufficient coverage in the initial library to detect them, or are they present, but not measurable?

This is addressed by Essential Revision #2.

3. In Figure 2b it would be beneficial to place the citation number of the paper corresponding to a previously published defect that matched your observations and border the matching mutation in the heatmap with a unique color identifier so that readers can assess the relative benchmarks and past known mutations and their effects in your assay format more easily.

This is addressed by Essential Revision #17.

4. For Figure S2 E and F the X-axis is the excitation laser rather than the emission spectra measured. As this signal is referred to as the emission miRPF 67o throughout the paper it may be helpful to change this label to the actual measured unit for the axis, as was done for the Y-axis (BV421).

This is addressed by Essential Revision #4.

5. For Figure S2 G, these appear to be a sub-sampled statistic of 1% of the total collected population. In the paper (line 634) it was noted that 2.1 million cells were collected, but only 1,000 of each population appear in this statistic table. Can you please also include the final statistics table to evaluate your final sort values. Can you also please note in the legend whether this data is a sub-sample of the published data, a benchmark run, or otherwise. If this gating format is not directly subsampled from the true data collection runs, please re-render these figures using the collected data.

This is addressed by Essential Revision #6.

6. Median fitness error to synonymous mutations is reported as "low" for the trafficking defect assay at 7.5%. It is again reported as low for the functional assays at 15.3%. It may be best instead to report the median and deviation of the synonymous and missense datasets and their relative differences at each tail end. Low is a subjective statement here and isn't supported by a positive control measure such as a large N of unmutated channels, run in the same sample, showing similar deviations in these assays as the synonymous mutants. If this data was run, it would be best to include it in the graph for comparison. While not essential, in the future I would encourage the authors to include unmutated controls into their library to assess the impact of synonymous substitutions, and to potentially make single synonymous substitutions at every amino acid position as a reference comparison to the mutants.

This is addressed by Essential Revision #7.

7. For Figure S3E-F, please show the FLIPR fluorescence of the miRFP 670 negative population as an overlay in F (dotted line of peaks over the red of P4) so we can gauge the fluorescence distribution of the negative control cells wherein no channel activity is present. Is this how "FLIPR-" was determined? Also, note in the figure the axis is labeled 640, which is the excitation laser. For consistency it may be best to label this axis as miRFP 670 emission rather than excitation.

This is addressed by Essential Revision #3.

8. The authors should include in their discussion some consideration of the "no data" category which generated a consistent number of analysis gaps within certain domains of the channel. How to avoid these in the future? Were these cloning issues? Library construction difficulties with SPINE? Poor coverage of the original cell pool? Could this have been avoided by making multiple cell libraries, sequencing them, and pooling those with compensating coverage?

This is addressed by Essential Revision #8.

9. The "no data" color scheme from Figure 6S should be adopted for the main paper figures. The use of light yellow for no data and white for "0 fold change" makes differentiating these two dataset types difficult visually in these figures. It is important for the reader to be able to easily observe where mutational coverage was lacking in this study.

This is addressed by Essential Revision #11.